# Single cell and spatial sequencing define processes by which keratinocytes and fibroblasts amplify inflammatory responses in psoriasis

Feiyang Ma [1,2,3], Olesya Plazyo[2], Allison C. Billi[2], Lam C. Tsoi[2], Xianying Xing[2], Rachael Wasikowski [2], Mehrnaz Gharaee-Kermani[2], Grace Hile[2], Yanyun Jiang[2], Paul W. Harms [2,4], Enze Xing [2], Joseph Kirma[2], Jingyue Xi[5], Jer-En Hsu[5], Mrinal K. Sarkar [2], Yutein Chung[2], Jeremy Di Domizio [6], Michel Gilliet [6], Nicole L. Ward [7], Emanual Maverakis [8], Eynav Klechevsky [9], John J. Voorhees [2], James T. Elder [2,10], Jun Hee Lee [5], J. Michelle Kahlenberg [1,2], Matteo Pellegrini[11], Robert L. Modlin [12,13,14] & Johann E. Gudjonsson [2,14] ✉

The immunopathogenesis of psoriasis, a common chronic inflammatory disease of the skin, is incompletely understood. Here we demonstrate, using a combination of single cell and spatial RNA sequencing, IL-36 dependent amplification of IL-17A and TNF inflammatory responses in the absence of neutrophil proteases, which primarily occur within the supraspinous layer of the psoriatic epidermis. We further show that a subset of *SFRP2*+ fibroblasts in psoriasis contribute to amplification of the immune network through transition to a pro-inflammatory state. The *SFRP2*+ fibroblast communication network involves production of *CCL13*, *CCL19* and *CXCL12*, connected by ligand-receptor interactions to other spatially proximate cell types: *CCR2*+ myeloid cells, *CCR7*+ *LAMP3*+ dendritic cells, and *CXCR4* expressed on both CD8+ Tc17 cells and keratinocytes, respectively. The *SFRP2*+ fibroblasts also express cathepsin S, further amplifying inflammatory responses by activating IL-36G in keratinocytes. These data provide an in-depth view of psoriasis pathogenesis, which expands our understanding of the critical cellular participants to include inflammatory fibroblasts and their cellular interactions.

Psoriasis is a common chronic inflammatory skin disease, affecting ~2% of the US population[1,2]. The etiology of psoriasis is complex. It has an incompletely understood but very significant environmental component[3], as well as a well-defined genetic predisposition, with over 60 genetic risk loci identified to date[4]. Psoriasis pathophysiology is also complex, with both autoimmune and autoinflammatory features playing key roles[5,6]. The result is a balance between T cell responses, dominated by the psoriasis-associated cytokines IFN-γ and IL-17, and innate immune-mediated responses, driven by high expression of IL-36[6,7]. While epidermal alterations and thickening in the context of keratinocyte hyperproliferation are the most obvious features of psoriasis, the disease is characterized by a much broader range of histologic alterations, including vascular proliferation and the influx of various inflammatory cell subsets including T cells, macrophages,

dendritic cells, and intraepidermal neutrophilic microabscesses[8]. However, research to date has primarily focused on a limited number of cellular players, including IL-17-producing CD4[+] (Th17) cells[9,10], myeloid-derived dendritic cells, macrophages[11], and keratinocytes[12]. In contrast, only limited attention has been paid to the pathogenic contributions attributable to other cell populations. Thus, a major knowledge gap remains in psoriasis pathogenesis: the lack of a comprehensive understanding of the different cellular components involved in psoriasis pathophysiology, including how they interact with one another.

Single-cell RNA-sequencing (scRNA-seq) has greatly expanded our abilities to gain novel insights into disease mechanisms. Several studies utilizing scRNA-seq analysis of psoriatic skin lesions have been performed[13–18]. These studies either mix psoriasis with other skin diseases to study disease heterogeneity or focus on specific cell types, such as CD8[+] T cells, emigrating cells, and myeloid cells in the epidermis. A major limitation of scRNA-seq analyses is that while these studies provide details regarding the individual cell types in the disease process, they cannot provide information on the spatial compartments in which cell–cell interactions occur at the site of disease.

Here, we present a comprehensive view of the immunopathogenesis of psoriasis using scRNA-seq of 33 biopsy samples derived from lesional, non-lesional, and healthy control skin. The data provide further insights into the pathogenesis of psoriasis, including (1) characterization of the cellular constituents participating in psoriasis pathogenesis, providing significant evidence that fibroblasts actively participate in psoriasis immunopathogenesis; (2) the interactions of the four cell types—fibroblasts, keratinocytes, T cells, and myeloid cells—as major drivers of cell–cell interactions in psoriasis skin; (3) localization of autoinflammatory responses within a specific epidermal compartment in psoriasis, and the function of IL-36G and its receptor IL-36R, in amplifying and sustain psoriatic inflammation; (4) association of psoriasis GWAS loci with distinct cellular populations; and (5) characterization of the major myeloid and T cell subsets contributing to psoriasis pathogenesis. Furthermore, using spatial sequencing, we provide additional in-depth information on the localization of the cell types and subtypes, as well as the localization of specific immune responses in psoriatic skin. These data present a more detailed view of psoriasis pathology, identifying further mechanisms by which cellular interactions contribute to the inflammatory network at the site of disease, which serve as potential targets for therapeutic intervention.

## Results

### Census of single cells from healthy and psoriatic human skin shows the diversity of skin cell populations

To understand the unbiased cellular composition and cell states of healthy and psoriatic human skin, we isolated single cell suspensions of skin cells from 8 healthy and 14 psoriasis biopsy specimens. We also collected skin cells peripheral to the lesions from 11 biopsy specimens from the same psoriasis patients, yielding a total of 33 10X Chromium single cell libraries (8 healthy skin libraries (NS), 11 perilesional psoriasis skin libraries (PN), and 14 lesional psoriasis libraries (PP)) (Supplementary Data 1). The resulting quality-controlled psoriasis single-cell atlas included 67,378 cells, with an average of 2421 genes and 10,953 transcripts detected per cell (Fig. 1a, b; Supplementary Data 2). To study the heterogeneity of these cells, we selected variable genes and performed uniform manifold approximation and projection (UMAP) dimensionality reduction and cell clustering using the R package Seurat[19]. Cluster annotation was corroborated by overlapping the cluster marker genes with canonical cell type-defining signature genes. We recovered 10 primary cell types including keratinocytes (KC; *KRT14*, *KRT1*, *DMKN*), melanocytes (MLNC; *DCT*, *TYRP1*, *PMEL*), eccrine gland cells (ECG; *PIP*, *DCD*, *MUCL1*), endothelial cells (EC; *PECAM1*, *CDH5*, *CLDN5*), fibroblasts (FB; *DCN*, *COL1A1*, *COL1A2*), smooth muscle cells (SMC; *ACTA2*, *TAGLN*, *MYL9*), nerve cells (Nerve; *MPZ*, *PLP1*,

*S100B*), T cells (TC; *CD3D*, *CD3E*, *TRAC*), myeloid cells (ML; *CD74*, *HLA-DRA*, *HLA-DPB1*), and mast cells (Mast; *CPA3*, *TPSAB1*, *CTSG*) (Fig. 1c; Supplementary Data 3).

The cell types we identified contained cells from the majority of NS, PN, and PP libraries, suggesting that each cell type was associated with a common cell lineage rather than derived from a single donor (Supplementary Fig. 1a). Strikingly, PP keratinocytes were completely separated from NS and PN keratinocytes in the UMAP, suggesting fundamental transcriptomic changes in psoriatic keratinocytes (Fig. 1b). By contrast, other cell types displayed overlapping patterns among the NS, PN, and PP cells. In terms of the number of cells obtained, the most abundant cell type was keratinocytes, followed by fibroblasts and T cells (Supplementary Fig. 1b). Analysis of the disease composition for each cell type showed an increased proportion of immune cells and endothelial cells in PP, with a decrease in eccrine gland cells, nerve cells, and melanocytes (Supplementary Fig. 1c). These results suggest that psoriasis is characterized by marked keratinocyte transcriptomic changes and accumulation of immune cells, including T cells and myeloid cells.

To localize the various cell types in the psoriasis skin site, we performed spatial-seq on the psoriasis skin. We collected RNA from a 20 µm-thick frozen section placed on top of an array of 55 µm wells containing spatially barcoded capture oligonucleotides for transcriptomic analysis (Supplementary Fig. 1d). After quality control steps, we detected 987 spatially defined spots with an average of 2782 genes and 11,779 transcripts per spot (Fig. 1d). We annotated the cell type composition for each spot by deconvoluting the spatial gene expression with the scRNA-seq gene expression of the major cell types (see the "Methods" section). The deconvolution results are displayed in a scatter-pie plot, which contains a pie chart for each spot in the spatial array (Supplementary Fig. 1e). *K*-means clustering revealed six clusters with distinct average cell type compositions, representing aggregates of keratinocytes, smooth muscle cells, fibroblasts, eccrine gland cells, endothelial cells, myeloid cells, and T cells (Fig. 1d; Supplementary Fig. 1f). The marker genes for each cluster in the spatial-seq were in agreement with the cell type marker genes determined by scRNA-seq, suggesting accurate annotations (Fig. 1e; Supplementary Data 4). As expected, keratinocytes were located in the epidermis. Myeloid cells, T cells, and endothelial cells were located in the superficial dermis in proximity to the epidermis, whereas fibroblasts, smooth muscle cells, and eccrine gland cells were located deeper in the dermis (Fig. 1d). We also observed a set of blank areas, within which low amount of mRNA was captured, representing the acellular area within the dermis. In parallel, we performed spatial-seq on three more psoriasis and two healthy skin biopsy specimens and observed similar cell types and spatial cellular aggregates (Supplementary Fig. 2; Supplementary Data 4).

### Distinct keratinocyte differentiation states in psoriatic skin reflect different cytokine responses in psoriasis

The distinct separation of PP from NS and PN keratinocytes suggests major shifts in epidermal function in lesional psoriatic skin (Fig. 1b). To characterize this further, we sub-clustered the keratinocytes and annotated the subtypes by differentiation state. Our analysis defined three distinct keratinocyte differentiation states within the basal layer (*COL17A1*, *DST*, *KRT15*), the spinous layer (*KRT6A*, *KRT6B*, *KRT16*), and the supraspinous layer (*SLURP1*, *KLK7*, *KRT2*) (Fig. 2a–c; Supplementary Data 5). We performed differential expression analysis between PP and NS keratinocytes within each epidermal differentiation state (basal, spinous, supraspinous) and identified the greatest number of differentially expressed genes (DEGs) in the supraspinous layer of psoriatic skin (Supplementary Fig. 3a; Supplementary Data 6). We plotted the top 15 up- and down-regulated genes for each differentiation state and found that PN keratinocytes, consistent across all three epidermal layers, exhibited an intermediate expression pattern

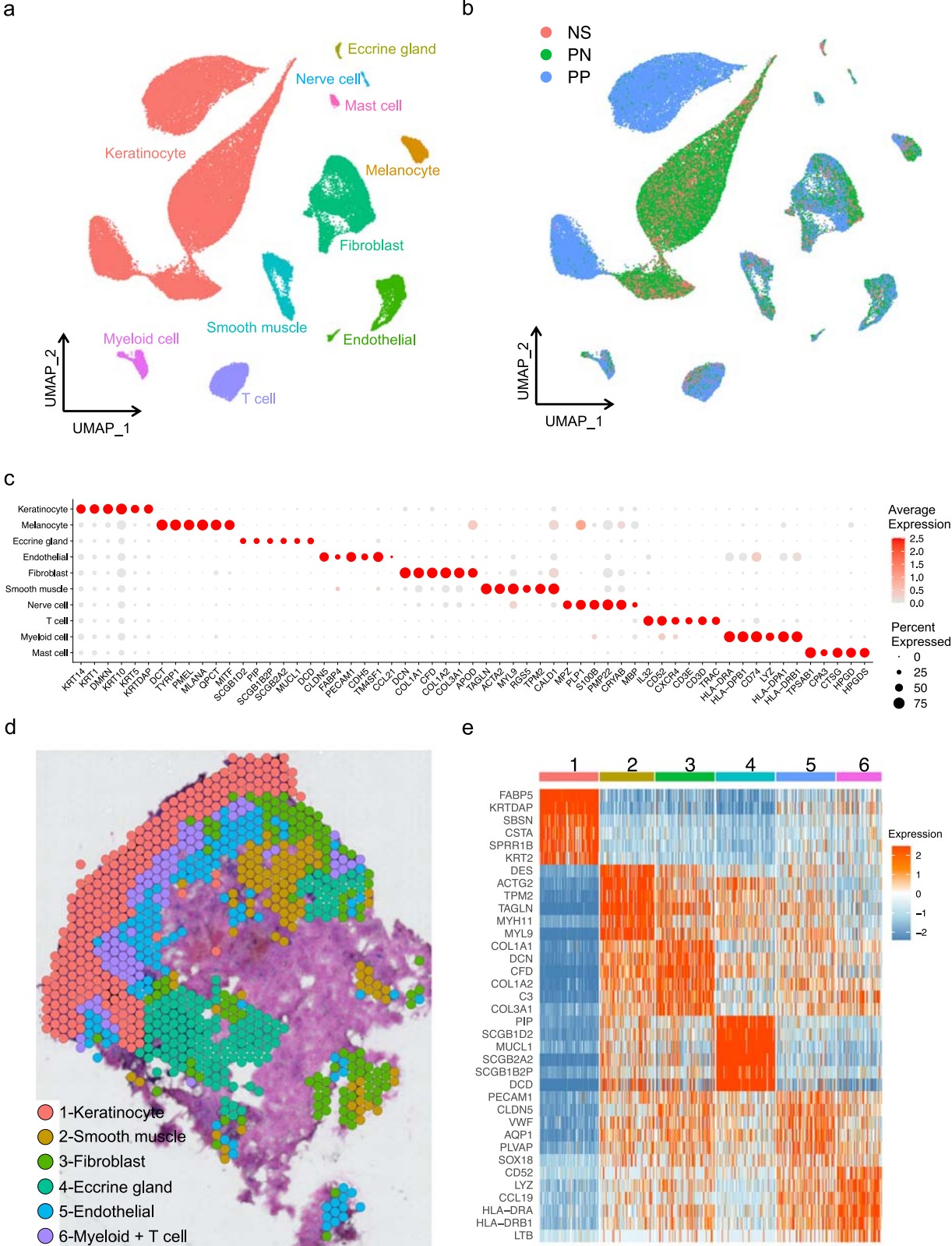

**Fig. 1 | Cell types observed in psoriasis skin and their spatial locations. a** UMAP plot showing 67,378 cells colored by cell types. **b** UMAP plot showing the cells colored by disease conditions. NS healthy normal skin, PN psoriatic non-lesional skin, PP psoriasis skin. **c** Dot plot showing representative marker genes for each cell type. The color scale represents the scaled expression average of each gene. The size of the dot represents the percentage of cells expressing each gene of interest. **d** Spatial plot for 764 spots colored by clusters, the coordinates of the spot correspond to the location in the tissue. **e** Heatmap showing the representative marker genes for each cluster in panel d. The color scale represents the scaled expression of each gene.

between that of NS and PP keratinocytes (Fig. 2d). *S100A7, S100A8,* and *S100A9* were markedly over-expressed in all three layers in PP, consistent with previous psoriasis studies[20]. Interestingly, type I interferon-induced genes, *IFITM3, IFI27,* and *IFI6* were highly ranked among the PP up-regulated genes, mostly localized to the basal epidermal layer in psoriatic skin. *KRT6* genes and *KRT16* were highly ranked in the spinous and supraspinous layers. Notably, *C1orf68*, previously identified as a target gene in psoriasis by genome-wide association studies (GWAS)[4], ranked first among PP down-regulated genes and was specifically expressed in the supraspinous layer (Fig. 2d).

We next used Ingenuity pathway analysis (IPA) to assess enriched pathways (Supplementary Fig. 3b–d) and cytokines that are upstream regulators for the DEGs within each epidermal layer (Supplementary Fig. 3e). This analysis implicated interferon signaling as the top mediator in both the basal and spinous layer, with a lesser function in the supraspinous layer in psoriatic skin (Supplementary Fig. 3b–d). The IL-17 pathway was predominately enriched in the supraspinous layer, with reduced enrichment in the spinous and basal layers. Other enriched upstream inflammatory responses in psoriatic skin included OSM, TNF, IL-36, IL-6, IL-1, IL-18, IL-21, IL-22, and IL-27 (Supplementary Fig. 3e).

To trace the source of these cytokines, we calculated the average expression of these genes in each cell type separated by the disease state (NS, PN, PP) (Supplementary Fig. 3f). The majority of these inflammatory mediators were expressed by a specific cell type, for example, *IL18* and *IL36G* by keratinocytes; *IL17A, TNF, IL21,* and *IL22* by T cells; and *IFNB1, IL1A, IL1B,* and *IL27* by myeloid cells. Notably, *IL36G, IL17A, IL21, IL22,* and *IL27* were most highly expressed in PP compared to PN and NS in the corresponding cell types. To validate the key upstream regulators, we calculated module scores using genes induced in cultured keratinocytes stimulated by individual pro-inflammatory cytokines, including type I IFN (IFN-α), IL-17A, IL-36γ, and TNF (Fig. 2e; Supplementary Data 7; see the "Methods" section). Consistent with the enrichment analysis on the DEGs, the IFN-α score increased from NS to PN to PP in all three layers, with the highest score in the basal layer, a slight decrease in the spinous layer, and a further decrease in the supraspinous layer. The IL-17A score was similar in NS and PN in all three layers but had a sharp increase from the basal to spinous to supraspinous layer in PP keratinocytes. Similarly, the IL-36γ score was higher in the supraspinous layer compared to the basal and spinous layers, in the PP epidermis. To validate the spatial specificity of these responses, we calculated the keratinocyte subtype module scores as well as IL-17A and IL-36γ module scores and projected these scores to 10 μm-sided square grids on a psoriasis skin biopsy specimen processed by Seq-Scope[21] (Supplementary Fig. 4a–d; see the "Methods" section). Consistent with the scRNA-seq results, both IL-17A and IL-36γ module scores were most highly correlated with the supraspinous compared to spinous or basal module scores (Supplementary Fig. 4e). These results suggest that distinct epidermal compartments are differentially influenced by pro-inflammatory cytokines in psoriasis.

To examine the role of IL-36 in activating IL-17 and TNF responses, we used CRISPR-Cas9 to knock out (KO) *IL36A, IL36G* and *IL1RL2* (IL-36R) in keratinocytes (Supplementary Fig. 3g) and measured the expression of four antimicrobial and proinflammatory genes induced by IL-17 and TNF: *DEFB4, S100A7, IL36G and IL36RN* (Fig. 2f). Deletion of either *IL36G* or *IL1RL2* led to a significant decrease in all four genes in keratinocytes, suggesting that IL-36G and IL-36R directly amplify IL-17A and TNF responses in the epidermis in the absence of neutrophil proteases (Fig. 2f). Notably, and in striking contrast, *IL36A* KO led to enhanced IL-17A and TNF responses as measured by mRNA expression of the antimicrobial genes *DEFB4* and *S100A7* (Fig. 2f).

Given the prominence of keratinocytes in psoriasis inflammatory responses, we sought to determine the ligand-receptor interaction among the three epidermal layers in psoriatic skin. To do this, we ran

CellPhoneDB[22] on all NS keratinocyte subtypes and all PP keratinocyte subtypes and analyzed the pairs showing higher interaction scores in PP compared to NS (see the "Methods" section). The most abundant ligand–receptor pairs among the three layers occurred between the ephrin family of ligands and their receptors, serving as a key signaling circuit in epidermal homeostasis[23] (Fig. 3a–c; Supplementary Data 8). Notch signaling interactions were also prominent among the keratinocyte layers and have been reported to play an essential function in epidermal growth and differentiation[24]. Interestingly, IL-36 signaling was only observed in the supraspinous layer in psoriatic skin, consistent with the scRNA-seq and spatial-seq expression data, which further assists leukocyte infiltration, particularly neutrophils, to the epidermis by inducing the chemokines *CXCL1* and *CXCL8* expressed by the supraspinous keratinocytes[25] (Fig. 3d; Supplementary Data 8).

To determine the spatial locations of the three keratinocyte subtypes, we sought to deconvolute the keratinocyte spots in spatial-seq using the subtype expression profiles defined by the scRNA-seq analyses (Supplementary Fig. 5a). The prediction scores accurately reflected the spatial location of the three layers of the epidermis, with supraspinous keratinocytes located in the outermost layer, spinous keratinocytes in the middle layer, and basal keratinocytes in the innermost layer (Supplementary Fig. 5a). We ran differential expression analysis between the PP keratinocyte spots and the NS keratinocyte spots and obtained a similar set of DEGs as from the scRNA-seq (Fig. 3e; Supplementary Data 9). Spatial mapping of two representative genes, *S100A7* and *IL36G*, illustrated their up-regulation in PP and specific location in the epidermis (Fig. 3f; Supplementary Fig. 5b), which was further confirmed by their expression in the scRNA-seq datasets (Supplementary Fig. 5c). Notably, *IL36G* was primarily detected in the spots predicted as supraspinous keratinocytes, consistent with the cytokine response results observed in our scRNA-seq analysis. *IL36G* was also detected in supraspinous keratinocytes by scRNA-seq of cells that emigrated from psoriasis skin biopsy specimens in vitro[15]. We also mapped the IL-17, IL-36, and TNF module scores on the spatial-seq samples and confirmed the co-localization of *IL36G* and *S100A7* expression with these cytokine responses (Fig. 3f).

The separation of the NS and PP keratinocytes in the UMAP suggests distinct keratinocyte differentiation pathways under healthy and disease conditions (Fig. 2b). To characterize these differentiation pathways, we performed pseudo-time analysis on NS and PP keratinocytes separately using Monocle[26]. Monocle pseudo time analysis arranged the cells into a linear trajectory, which followed the direction of basal to spinous to supraspinous for both NS (Supplementary Fig. 6a) and PP (Supplementary Fig. 6b) keratinocytes. To identify the potential cytokines that drive the differentiation, we split the variable genes along the pseudo time into five expression patterns for both NS (Supplementary Fig. 6c; Supplementary Data 10) and PP (Supplementary Fig. 6d; Supplementary Data 10) trajectories. Next, we inferred the upstream regulators for the genes in each expression pattern using IPA. For each upstream regulator, we calculated a module score using all target genes across the five expression patterns. We then calculated the correlation between the module scores for each upstream regulator and the pseudotime defined by the Monocle analysis. We found that only OSM and IL-4 module scores were positively correlated with NS keratinocyte pseudotime, while the module scores for multiple cytokines, including IL-22, IL-1β, IL-17A, IL-13, and TNF, were highly correlated with PP keratinocyte pseudotime (Supplementary Fig. 6e). To validate the key cytokines driving PP keratinocyte differentiation, we calculated module scores using genes induced in cultured keratinocytes stimulated by individual cytokines, including IL-1β, IL-17A, IL-13, and TNF (Supplementary Fig. 6f; Supplementary Data 7). The module scores for these four cytokines were highly correlated with the PP keratinocyte pseudotime, consistent with the results inferred by IPA. These results suggest that keratinocyte differentiation in psoriasis, but not normal skin, is driven by the local

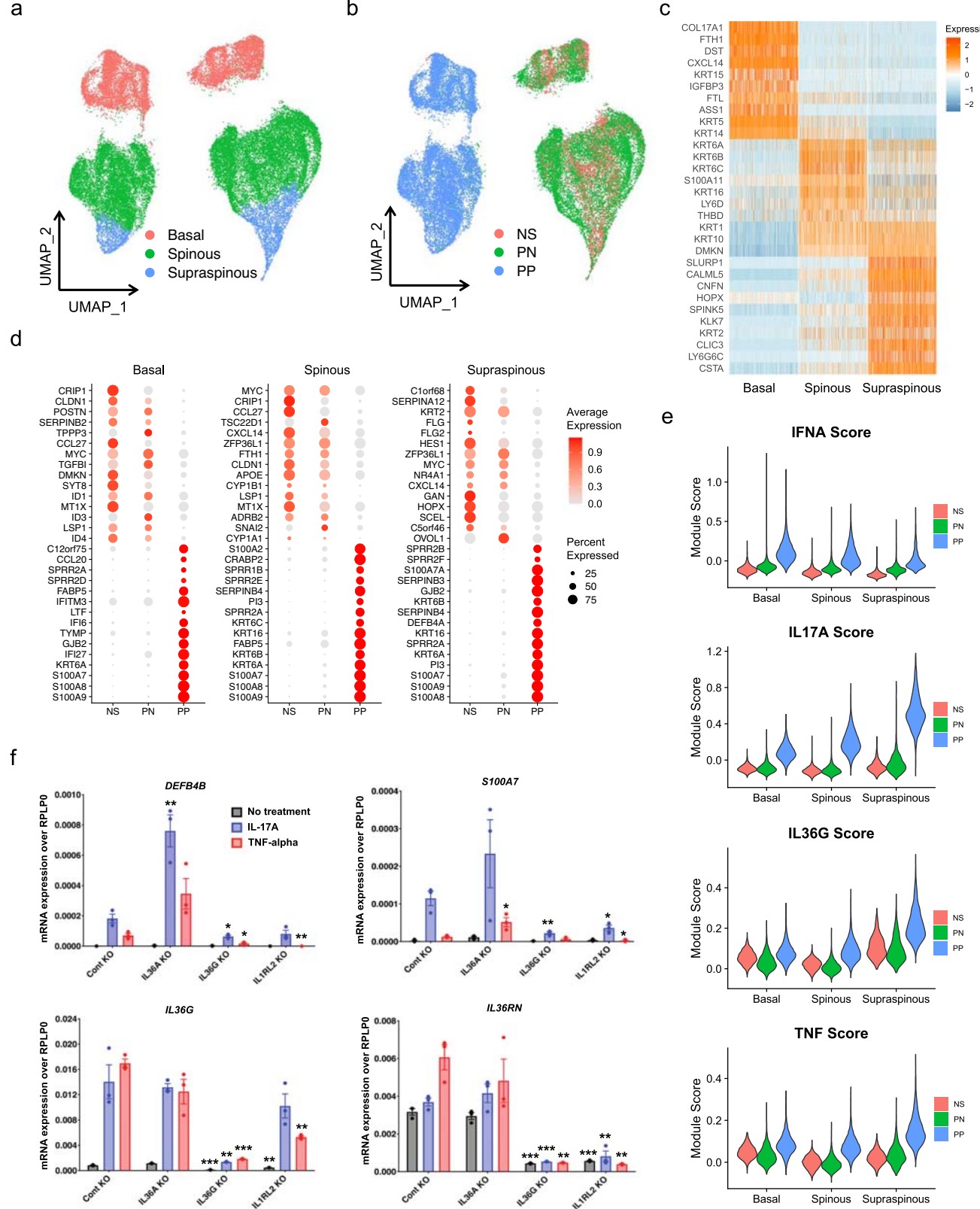

**Fig. 2 | IL-36 amplifies IL-17 response in the supraspinous epidermis of psoriatic skin. a** UMAP plot showing 41,923 keratinocytes colored by cell types. **b** UMAP plot showing the keratinocytes colored by disease conditions. **c** Heatmap showing marker genes with the highest fold change for each subtype. The color scale represents the scaled expression of each gene. **d** Dot plot showing the top 15 differentially expressed genes comparing PP to NS in the basal (left), spinous (middle), and supraspinous (right) layers. The color scale represents the scaled expression of each gene. The size of the dot represents the percentage of cells expressing each gene of interest. **e** Violin plot showing the cytokine module scores in the keratinocyte subtypes, and each subtype is split by the disease conditions. **f** qRT-PCR of four genes in *IL36A*, *IL36G* and *IL1RL2* KO keratinocytes after treatment with IL-17A and TNF (*n* = 3 biologically independent; unpaired *t*-test; *$P < 0.05$, **$P < 0.01$, ***$P < 0.001$; mean ± SEM).

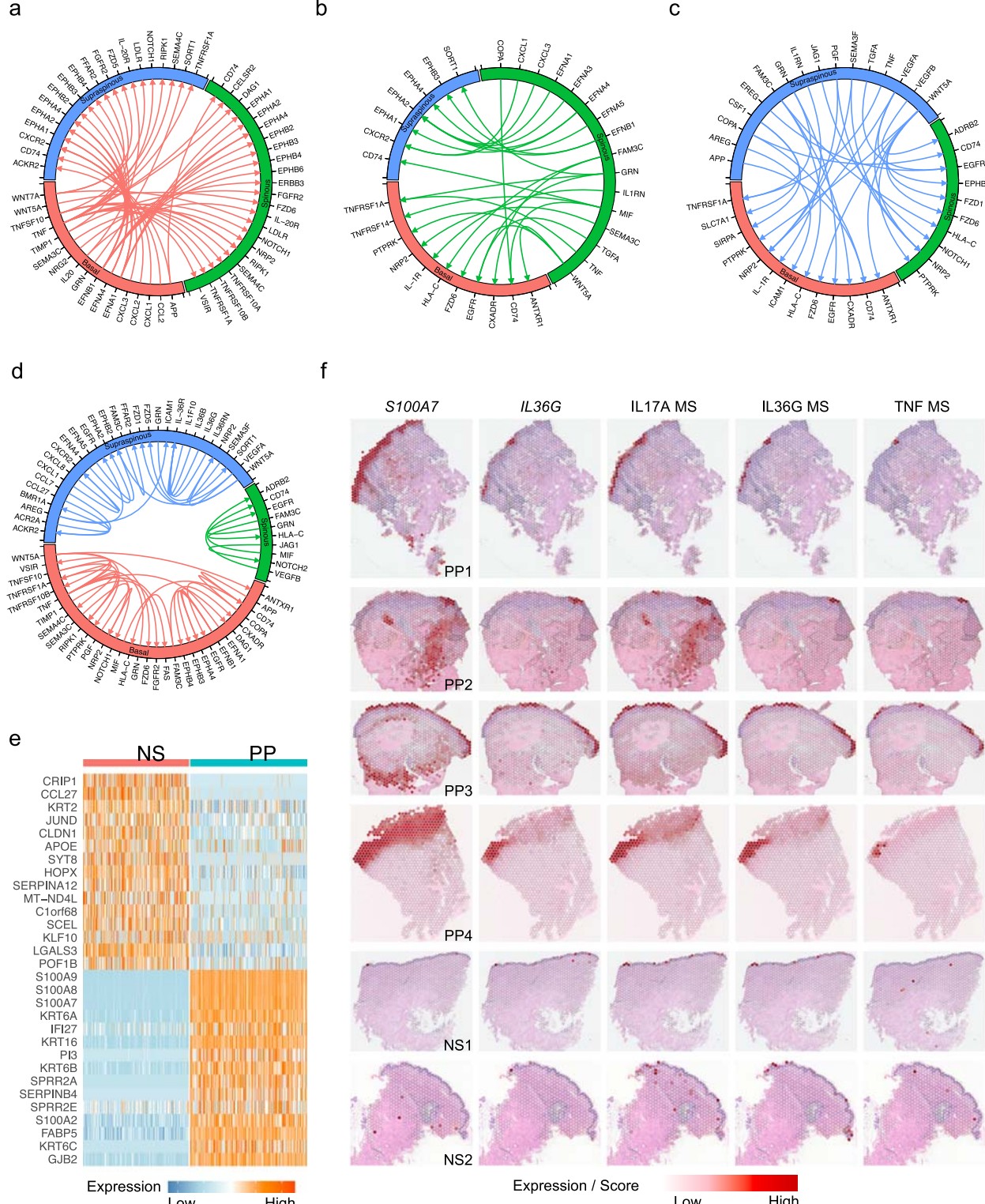

**Fig. 3 | Ligand–receptor interaction and spatial location of the keratinocyte subtypes.** **a** Circos plot showing the ligand–receptor interactions with a higher score in PP compared to NS. The ligands are expressed by the basal keratinocytes, the receptors are expressed by the other subtypes. **b** Circos plot showing the ligand–receptor interactions with a higher score in PP compared to NS. The ligands are expressed by the spinous keratinocytes, the receptors are expressed by the other subtypes. **c** Circos plot showing the ligand–receptor interactions with a higher score in PP compared to NS. The ligands are expressed by the supraspinous keratinocytes, the receptors are expressed by the other subtypes. **d** Circos plot showing the ligand-receptor interactions with a higher score in PP compared to NS. The ligands and receptors are expressed by the same subtype. **e** Heatmap showing the top differentially expressed genes between NS and PP keratinocytes identified in both scRNA-seq and spatial-seq. **f** Spatial plots showing expression level of *S100A7* and *IL36G*, and the downstream target module scores for IL-17A, IL-36G, and TNF.

production of multiple cytokines, particularly IL-1β, IL-22, IL-17A, IL-13, and TNF.

Keratinocyte hyperproliferation is a hallmark feature of psoriasis. To address keratinocyte proliferation, we studied the cell cycle effect across NS, PN, and PP (Supplementary Fig. 7a–d), by adding back proliferating keratinocytes, which were excluded from our initial assessment to facilitate analyses of cellular heterogeneity. Expectedly, more than 50% of the proliferating keratinocytes were derived from the PP samples compared to NS or PN samples (Supplementary Fig. 7e). We mapped the cell cycle score on the spatial-seq samples and identified the strongest proliferation effect in the epidermis, particularly in the basal compartment of the epidermis (Supplementary Fig. 7f). We also revealed a thicker layer of proliferating keratinocytes in the PP samples compared to the NS samples, confirming the hyperproliferation phenotype in PP in our spatial-seq data.

## Contribution of *SFRP2*+ fibroblasts to psoriasis inflammation is distinct from their pro-fibrotic role

Given that fibroblast were the second most frequent cell type in psoriasis lesions as defined by the scRNA-seq data, we investigated the contribution of fibroblasts to the disease. We sub-clustered all fibroblasts into 12 sub-clusters (Fig. 4a–c; Supplementary Data 11). Based on previously published skin fibroblast marker genes[27], we annotated the sub-clusters into four subtypes: *SFRP2*+, *COL11A1*+, *SFRP4*+, and *RAMP1*+ fibroblasts (Fig. 4b; Supplementary Fig. 8a, b). We confirmed the presence of the major fibroblast subtypes by immunohistochemistry (IHC) in lesional psoriatic skin (Supplementary Fig. 10a). *SFRP2*+ fibroblasts were the largest fibroblast subpopulation in the skin, consistent with previous studies[27]. Although *SFRP2*+ fibroblasts contained cells from PP, PN, and NS, notably, PP fibroblasts were mostly separated from PN and NS fibroblasts by dimensionality reduction using UMAP (Fig. 4c). To define the mechanisms that drive this shift in psoriatic fibroblasts, we identified DEGs between PP and PN fibroblasts and performed enrichment analysis using IPA (Supplementary Data 12). The activated upstream regulator analysis identified type I IFNs, IFN-γ and TNF as the top mediators promoting psoriasis-associated changes in the *SFRP2*+ fibroblast subpopulation (Fig. 4d). Interestingly, we found these upstream regulators were also activated in the three keratinocyte subtypes by comparing PP to PN cells (Supplementary Fig. 8c; Supplementary Data 12), suggesting that these are common drivers that shift both fibroblasts and keratinocytes from PN to PP.

Within the *SFRP2*+ fibroblasts, there were four sub-clusters primarily composed of PP fibroblasts, sub-cluster 3, 0, 6, and 2 (Fig. 4c, e). Interestingly, differential expression analysis suggested substantial heterogeneity in PP *SFRP2*+ fibroblasts, with collagen genes (e.g., *COL1A1* and *COL3A1*) identified in sub-cluster 3 markers and a set of pro-inflammatory cytokines (e.g., *CCL19, IL33, IL34*), as well as the protease *CTSS* in sub-cluster 6 markers (Fig. 4f; Supplementary Data 13). These chemokines and proteases were most prominently expressed in fibroblasts in the upper dermis and at the tips of the dermal papillae as revealed by both spatial-seq mapping (Supplementary Fig. 9a) and immunofluorescence (Fig. 4g; Supplementary Fig. 9b). Analysis of the upstream regulators for fibroblast sub-cluster 3 and sub-cluster 6 marker genes showed the greatest increase of the activation z scores for key psoriasis-associated cytokines IFN-γ, TNF, IL-1β, OSM, and IL-17A, with a decrease for IL-5, EDN1, and IL-4, all reported to be associated with fibrosis (Fig. 4h)[28–30]. Taken together, these results reveal heterogeneity in PP *SFRP2*+ fibroblasts that include producers of extracellular matrix components, such as type I collagen, as well as cytokine-induced, non-matrix producing, pro-inflammatory compartments.

## CD8+ Tc17 is a major source of IL17A in psoriatic skin

IL-17-producing T cells have long been considered the central player orchestrating psoriasis pathogenesis[31,32]. To define the roles of the

T cell subtypes in psoriasis, we sub-clustered the T cells and annotated the sub-clusters using canonical T cell lineage markers (Fig. 5a, b). We identified seven T cell subtypes in NS, PN, and PP skin: naïve T cells (sub-clusters 0 and 7; *RGCC, CCR7, IL7R*), stressed T cells (sub-clusters 1 and 2; *DNAJB1, HSPA1A, HSPH1*), regulatory T cells (Tregs) (sub-cluster 3; *FOXP3, TIGIT, BAFT*), cytotoxic T cells (sub-cluster 4; *CCL5, NKG7, GZMK*), IL17-secreting CD8 T (Tc17) cells (sub-cluster 5; *IL17A, IL17F, CCL20*), gamma delta T (γδ T) cells (sub-cluster 6; *TRDC, FCER1G, TYROBP*), and IFN T cells (sub-cluster 8; *IFI6, MX1, IFIT3*) (Supplementary Fig. 11a; Supplementary Data 14). Notably, most of the T cells were derived from the PP samples, consistent with the known influx of T cells in psoriatic skin (Fig. 5b). Analysis of the disease composition for each sub-cluster revealed an increased proportion of Tregs, Tc17 cells, and IFN T cells in PP (Fig. 5c). Consistent with the findings from keratinocytes, the increased proportion of T cells manifesting an IFN signature in PP indicates that the strong IFN response in the psoriatic skin sites impacts not only keratinocytes but also other cell types. We examined the expression of cytokines previously reported to be involved in psoriasis, and found that *IL17A, IL17F, IL26, CCL20, CXCL13, IL22*, and *IL23R* were specifically expressed by Tc17 cells (Supplementary Fig. 11b). IL-17 producing T cells have historically been mostly considered to be CD4+ T cells[10], although recent observations have provided growing evidence for IL-17 producing CD8+ T cells in psoriasis[33–35]. The identification of both CD4+ and CD8+ T cells from the skin enables direct comparison of these T cell subsets and their contribution to IL-17 response in psoriatic skin. Here, our data suggest that CD8+ T cells are a major source of IL17A in psoriasis. To further characterize this, we inspected the co-expression of *IL17A* with *CD4, CD8A*, and *IFNG* and found a positive correlation between *IL17A*-expressing cells and *CD8A* and *IFNG*-expressing cells, yet few *CD4+ IL17A+* cells were observed (Fig. 5d, Supplementary Figs. 11c, 12a–e). In addition, our data demonstrate that there is a higher number of *IL17A* than *IL17F*-producing T cells, although the majority were positive for both (Fig. 5f). We mapped the T cell marker genes in the spatial-seq samples, and revealed that, although few IL-17 or TNF producing T cells were identified in the epidermis, most T cells were detected at the dermal-epidermal junction (Fig. 5e), which was further validated by immunofluorescence (Supplementary Fig. 11d). Interestingly, we found that IL-17 receptor gene (*IL17RA*) and TNF receptor gene (*TNFRSF1A*) were highly expressed in the epidermis of the PP samples (Fig. 5e), consistent with the localized cytokine responses identified in keratinocytes[36].

## Single-cell RNA sequencing defines unique myeloid subsets in psoriasis

To examine cellular heterogeneity in myeloid cells in psoriatic skin, we sub-clustered myeloid cells and annotated six myeloid subpopulations: Langerhans cells (LC; *CD207* and *CD1A*), classical type 1 dendritic cells (cDC1; *CLEC9A* and *IRF8*), classical type 2 dendritic cells subset A (cDC2A; *LAMP3* and *CD1B*), classical type 2 dendritic cells subset B (cDC2B; *CLEC10A* and *IL1B*), CD16+ dendritic cells (CD16+ DC; *FCGR3A* and *HES1*), and perivascular macrophages (PVM; *CD163* and *SELENOP*) (Fig. 6a, b; Supplementary Data 15). Analysis of the disease composition for each subtype revealed a decreased proportion of LCs in PP (Fig. 6c). Previous studies have suggested impairment of LC migration and function in psoriatic skin[37], and our analyses were consistent with this, revealing diminished numbers of LC in PP[38]. Strikingly, cDC2A was exclusively derived from PN and PP skin, with greater numbers found in PP skin. Previous studies have reported increased expression of *IL15* and *IL32* in psoriasis, both cytokines serve as proinflammatory mediators[39,40] and were mostly expressed by cDC2A myeloid cells (Fig. 6d). We also found that *CCL17, CCL19*, and its receptor, *CCR7*, were primarily expressed by cDC2A (Fig. 6d). Inhibition of the CCL19/CCR7 axis in lesional skin has been reported to be a critical event for clinical remission induced by TNF blockade in patients with psoriasis[41].

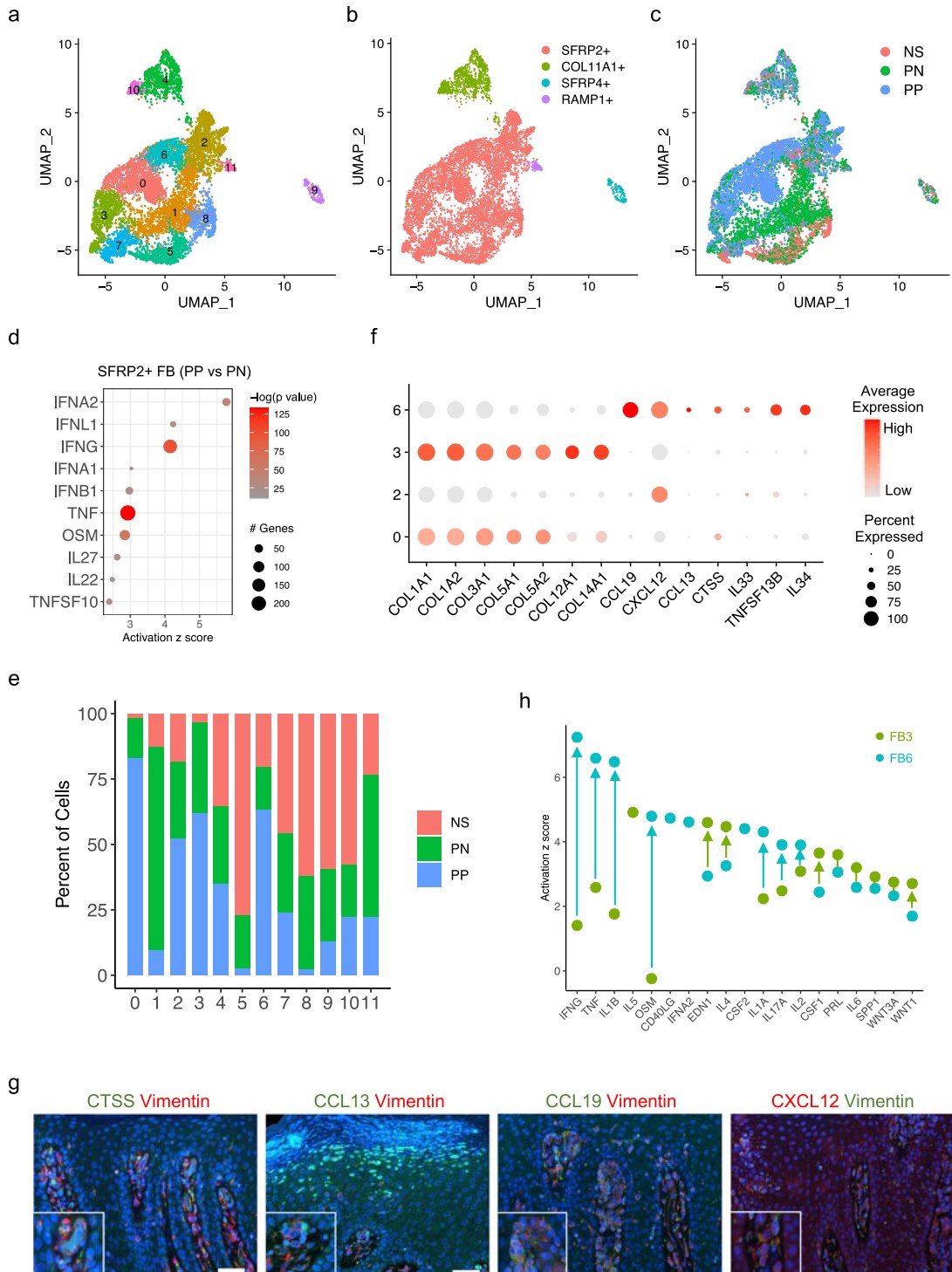

**Fig. 4 | *SFRP2*+ fibroblasts transition from pro-fibrotic state to pro-inflammatory state in psoriatic skin. a** UMAP plot showing 9327 fibroblasts colored by sub-clusters. **b** UMAP plot showing the fibroblasts colored by subtypes. **c** UMAP plot showing the fibroblasts colored by disease conditions. **d** Dot plot showing the top 10 cytokine upstream regulators for differentially expressed genes comparing PP to PN in the *SFRP2*+ fibroblasts. The color scale represents the −log10(*p* value) from the enrichment analysis. The size of the dot represents the number of differentially expressed genes downstream of each upstream regulator. Hypergeometric test was used for the enrichment analysis, and the Benjamini−Hochberg procedure was used for false discovery rate adjustment. **e** Bar plot showing the abundance composition across the disease conditions for each fibroblast sub-clusters. **f** Dot plot showing collagen marker genes for sub-cluster 3 and inflammatory marker genes for sub-cluster 3. The color scale represents the scaled expression average of each gene. The size of the dot represents the percentage of cells expressing each gene of interest. **g** Immunofluorescence showing the colocalization of CTSS, CCL13, CCL19, and CXCL12 with Vimentin in the dermal papillae. **h** Scatter plot showing the top upstream regulators for marker genes in PP fibroblast cluster 3 and cluster 6. The color represents the cluster identity of the upstream regulator. The arrows show the change of the activation z scores.

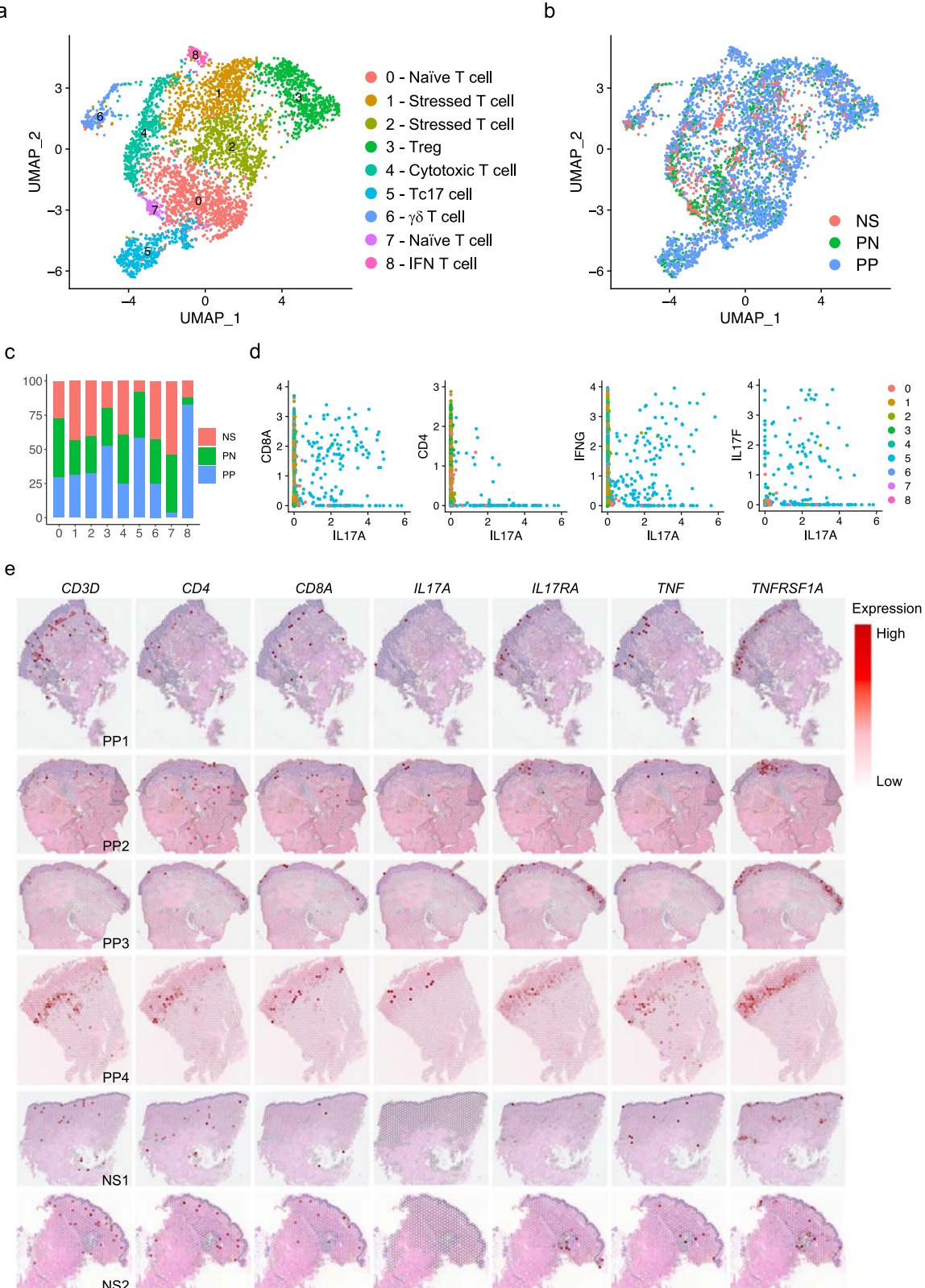

**Fig. 5 | CD8⁺ Tc17 is a major source of *IL17A* in psoriatic skin. a** UMAP plot showing 4468T cells colored by sub-clusters. **b** UMAP plot showing the T cells colored by disease conditions. **c** Bar plot showing the abundance composition across the disease conditions for each sub-cluster. **d** Scatter plot showing co- expression of pairs of genes. The color represents the sub-cluster identity of the cells. **e** Spatial plots showing the expression level of *CD3D, CD4, CD8A, IL17A, IL17RA, TNF,* and *TNFRSF1A* in all the spatial-seq samples.

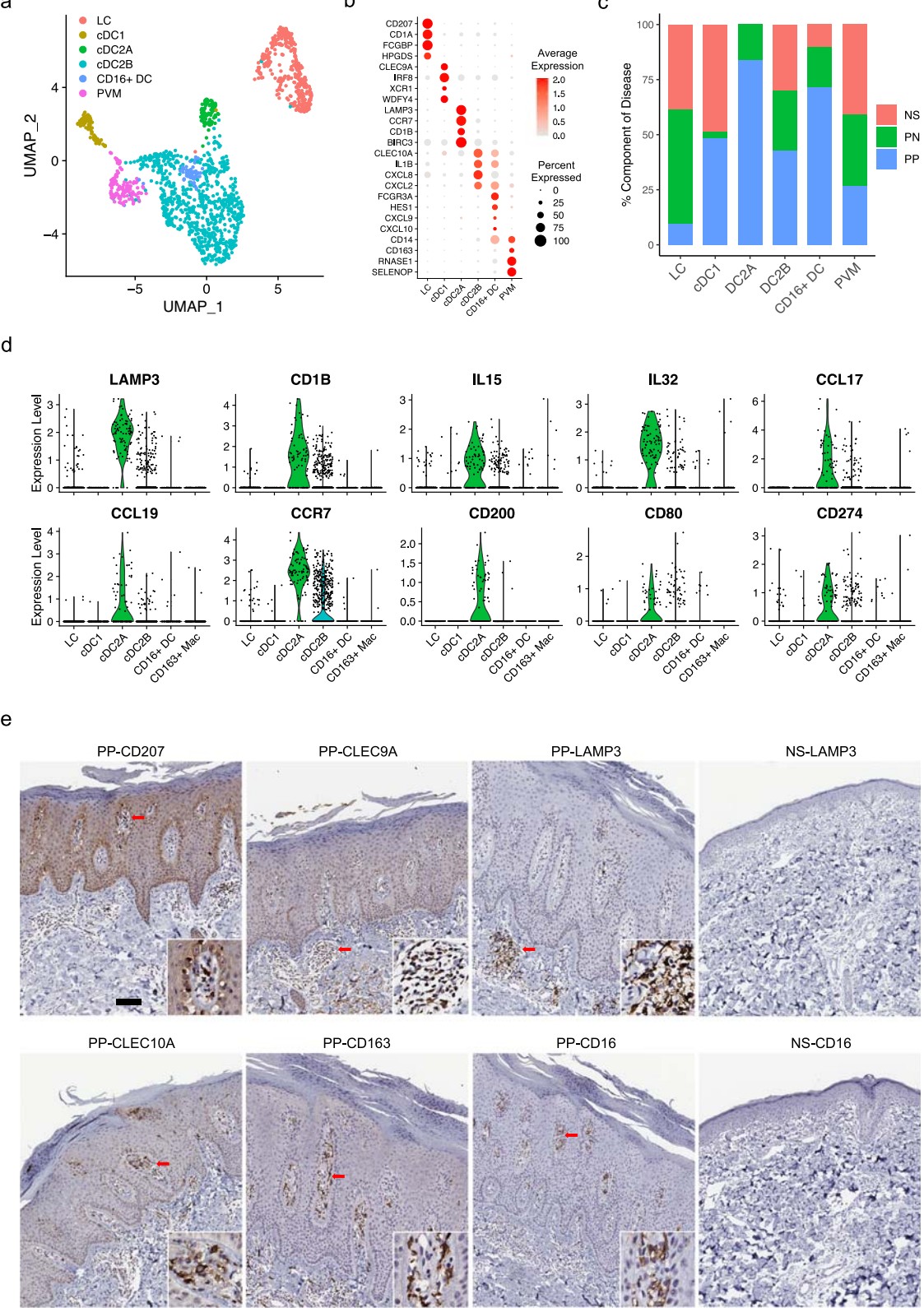

**Fig. 6 | Single-cell RNA sequencing defines unique myeloid subsets in psoriasis.** **a** UMAP plot showing 1436 myeloid cells colored by subtypes. **b** Dot plot showing the representative marker genes for each subtype. The color scale represents the scaled expression of the gene. The size of the dot represents the percentage of cells expressing the gene of interest. **c** Bar plot showing the abundance composition across the disease conditions for each subtype. **d** Violin plot showing the expression of genes split by subtype. Each dot represents the gene's expression in a single cell. **e** Immunohistochemistry staining for marker genes for each myeloid subtype. LAMP3 and CD16 were shown in both NS and PP samples. The red arrow points to the zoom-in location. The size bar represents 100 μm.

Furthermore, cDC2A expressed high levels of the co-stimulatory molecules *CD200*, *CD80*, and *CD274*, suggesting that these cells may prime and drive T cell activity and expansion in psoriatic skin (Fig. 6d). CD16[+] DC were predominantly derived from PP samples (Fig. 6c). CD16[+] DC had a similar expression pattern as cDC2B, including *CLEC10A*, *IL1B*, and *CXCL2*, but also expressed *FCGR3A* (CD16), *CD14*, and *HES1*, suggesting their origin from circulating CD14[+] CD16[+] monocytes[42] (Fig. 6b). Notably, we found that *CXCL9* and *CXCL10* were mainly expressed by CD16[+] DC, these two chemokines have been shown to drive effector Th1/Th17 cell polarization via STAT1, STAT4, and STAT5 activation[43] (Fig. 6b). These data were validated by IHC in psoriatic and normal skin and showed different localization of these subsets in psoriatic skin, with CLEC9A and LAMP3 predominantly localized in lymphoid like structures in the dermis and CLEC10A and CD16 predominantly within the tips of the dermal papillae (Fig. 6e). LAMP3[+] and CD16[+] cells were not found in normal skin. PVM was found in the perivascular location in the dermal papillae, whereas LCs were found primarily along the dermal–epidermal junction in the dermal papillae (Fig. 6e).

### Single-cell RNA sequencing highlights cell type specificity of psoriasis-associated genetic risk variants

To determine the genes associated with the psoriasis-related genetic variants and their cellular locations, we examined 65 single nucleotide polymorphism (SNP) loci detected by GWAS in psoriasis[4]. A total of 46 genes associated with these SNPs were identified using an expression quantitative trait loci (eQTL) database for human skin (see the "Methods" section). To increase the number of SNP-related genes, we also considered the nearest gene to SNP loci in the human genome, which yielded 65 associated genes. After filtering out genes with low expression, we obtained 27 eQTL-associated genes and 57 nearest genes. We calculated the average expression of these genes in each cell type separated by NS, PN, and PP and identified the cell type that expressed each of these genes (Supplementary Fig. 13a, b). Notable results include the demonstration of *C1orf68* expression primarily in keratinocytes, with a decreasing pattern from NS to PN to PP. *IL23R*, which has been shown to promote an IL-17 response in psoriasis[44], was primarily expressed in T cells from PN and PP. These results provide a valuable resource for functional studies aimed at understanding the mechanisms of psoriasis-associated genetic loci and the critical cell populations involved.

### Ligand–receptor analysis shows cell type-specific networks in psoriasis

Given the observed shifts in cell type and subtype composition of psoriatic skin compared to healthy skin, we analyzed how cell–cell communication changes with the development of psoriasis. To do this, we first further sub-clustered the endothelial cells, smooth muscle cells, and melanocytes (Supplementary Fig. 14a–c; Supplementary Data 16). We then performed separate ligand–receptor analyses on the NS-specific sub-clusters and the PP-specific sub-clusters using CellPhoneDB (see the "Methods" section). To assess the changes that occur in psoriasis, we analyzed ligand–receptor pairs that had higher interaction scores in PP compared to NS. The greatest number of ligand–receptor pair changes in psoriatic skin was seen for four cell types: keratinocytes, fibroblasts, myeloid cells, and T cells (Fig. 7a; Supplementary Data 17). We reasoned that adjacency in spatial location would facilitate cell–cell communication. To this end, we counted the number of neighboring spots for each pair of cell types as shown in the scatter-pie plot (Fig. 7b; see the "Methods" section). The results indicate that keratinocytes had the most immune cell neighbors (T cells and myeloid cells), fibroblasts were spread across many other cell types, and the endothelial cells were adjacent to immune cells (Fig. 7c). By aggregating the number of ligand–receptor pair changes and the spatial adjacency analysis,

we prioritized our analyses on keratinocyte, fibroblast, myeloid, and T cell interactions.

Notably, predicted interactions from keratinocytes include *CCL2* and *CCL7* to *CCR2* in myeloid cells, *CCL20* to *CCR6* and *IL7* to *IL7R* in myeloid cells and T cells, and insulin-like growth factor family ligands (*IGFL1*, *IGFL2*, *IGFL3*) to their receptor (*IGFLR1*) on T cells (Fig. 7d), suggesting that keratinocytes regulate the immune response in psoriasis. In addition, keratinocytes expressed platelet-derived growth factor (*PDGFA*, *PDGFC*), transforming growth factors (*TGFB1*), epidermal growth factor (*TGFA*), and vascular endothelial growth factor (*VEGFA*, *VEGFB*) capable of interacting with their respective receptors in fibroblasts (Fig. 7d).

Surprisingly, fibroblasts were connected to many cell types through putative ligand–receptor interactions. Fibroblasts produced a number of chemokines (*CCL13*, *CCL19*, *CCL2*, *CCL8*, *CXCL1*, *CXCL12*, *CXCL2*, *CXCL3*) that linked to receptors expressed by myeloid cells and T cells, suggesting an important role for fibroblasts in recruiting immune cells in psoriasis (Fig. 7e). Fibroblasts were also the source of the cytokines IL-7 and IL-15 that link to the IL-7 receptor and the IL-15 receptor on T cells, promoting T cell proliferation and maturation[45,46]. Fibroblasts have the capacity to induce hyperproliferation of keratinocytes by expression of fibroblast growth factor (*FGF10*, *FGF2*, *FGF7*) coupled with their receptors, which we detected on keratinocytes[47–49]. We validated the expression of the fibroblast growth factors FGF2 and FGF7 and their receptors FGFR2 and FGFR3 by IHC in lesional psoriatic skin (Supplementary Fig. 10b).

Myeloid cells expressed multiple chemokines that further promote additional leukocyte recruitment to psoriatic sites (Fig. 7f). These chemokines also interact with the atypical chemokine receptors (*ACKR2*, *ACKR3*, *ACKR4*) expressed by both keratinocytes and fibroblasts, provoking excessive Th17 responses and amplifying skin inflammation[50]. Furthermore, myeloid cells expressed cytokines (*IL15*, *IL1A*, *IL1B*, *IL6*) that inhibit apoptosis in keratinocytes[51]. T cells expressed cytokines (*CCL3*, *CCL3L1*, *CCL4*, *CCL5*, *CXCL13*) that interact with atypical chemokine receptors expressed by keratinocytes and fibroblasts (Fig. 7g). In addition, T cells expressed *IL17A*, *IL17F*, and *IL22* that, coupled with their receptors expressed by myeloid cells and keratinocytes, are known to promote inflammation in psoriasis.

Thus, by integrating the scRNA-seq data, upstream regulator analysis, the ligand–receptor results, and the spatial locations of different cell types, we were able to create a model that highlights the role of fibroblasts in psoriasis skin (Supplementary Fig. 15a, b). In psoriasis, *SFRP2*[+] fibroblasts transition from a fibrotic to an inflammatory state. These fibroblasts expressed *CCL13* and *CCL19*, able to recruit the relevant chemokine receptor-expressing cells: *CCR2*[+] myeloid cells and *CCR7*[+] *LAMP3*[+] cDC2A, respectively. These inflammatory fibroblasts also expressed high levels of *CXCL1*, known to recruit neutrophils, and *CXCL12*, known to recruit *CXCR4*[+] Tc17. *CXCL12* can also act on the epidermis via its receptor *CXCR4* to induce keratinocyte proliferation. In the epidermis, IL-17 secreted by the Tc17 induces inflammation, further amplified by the IL-36 loop within the supraspinous layer, triggering the release of *CCL20* which can act to recruit *CCR6*[+] Tc17 as well as *CXCL1* and *CXCL8* which can recruit neutrophils, consistent with the microabscesses in the supraspinous epidermal layer.

## Discussion

The data presented here identify a cell–cell interaction network in the pathogenesis of psoriasis, highlighting the prominent contribution of stromal cells to epithelial cell dysregulation. Specifically, using a combination of single cell and spatial RNA sequencing, we identify that an *SFRP2*[+] fibroblast subset, via production of *CCL13*, *CCL19*, *CXCL1*, and *CXCL12*, impacts other spatially proximate cell types: *CCR2*[+] myeloid cells, *CCR7*[+] *LAMP3*[+] dendritic cells, neutrophils, as well as *CXCR4*[+] CD8[+] Tc17 cells and *CXCR4*[+] keratinocytes, respectively. The action of fibroblasts both directly via *CTSS* and indirectly via CD8[+] Tc17 cells

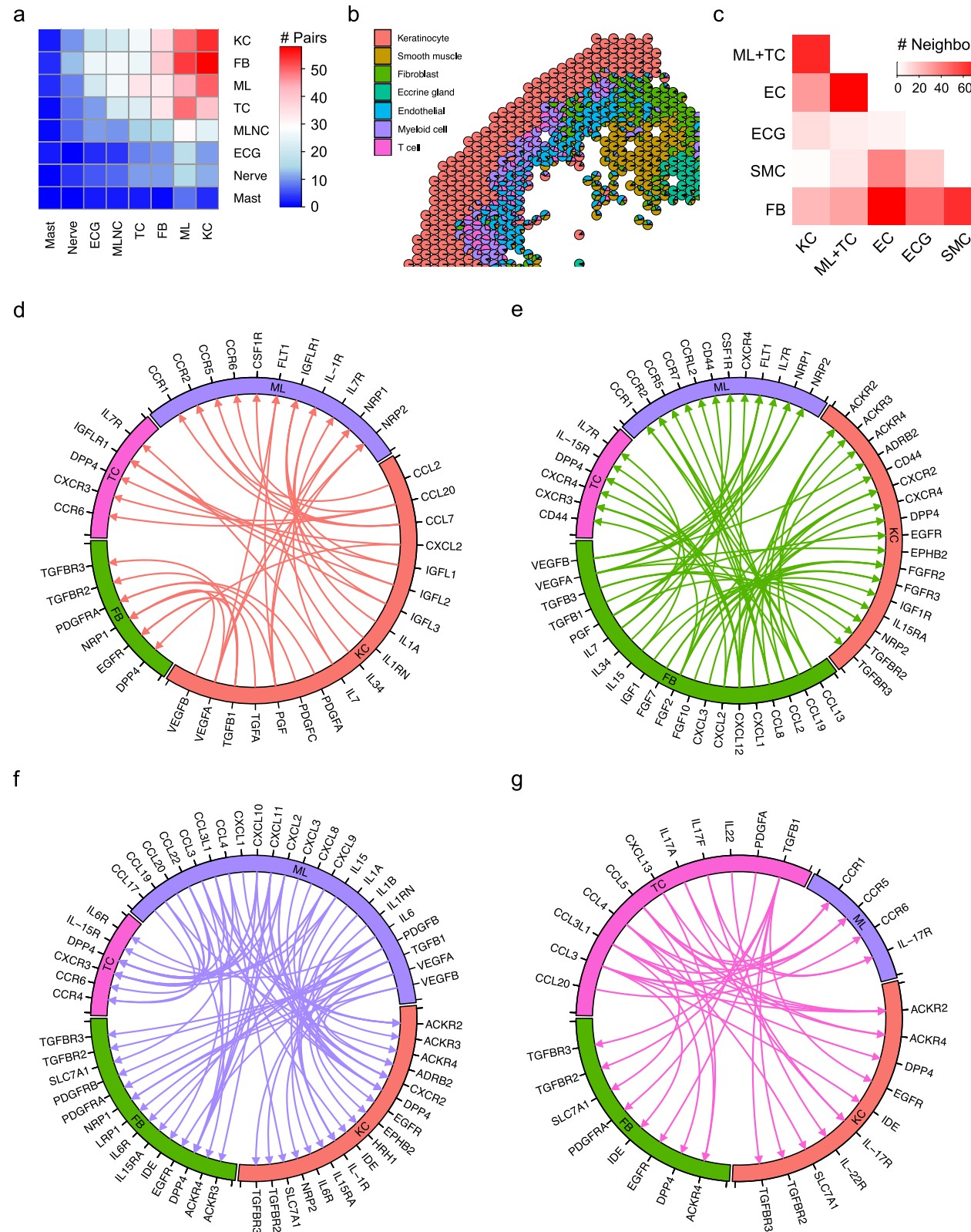

production of IL-17 results in a striking IL-36-driven amplification loop compartmentalized within the supraspinous layer of the psoriatic epidermis, that in the absence of neutrophil proteases augments IL-17A and TNF inflammatory responses.

Our data provide detailed insights into the role of fibroblasts, an underappreciated contributor to psoriasis pathogenesis. Prior studies on the role of fibroblasts have mostly been limited to the ability of

fibroblasts to induce keratinocyte proliferation[52–54] and immune responses[54], such as via secretion of fibroblast growth factors such as KGF[49], FGF10[55], or CXCL12[54]. Interestingly, we observed heterogeneous subpopulations in PP *SFRP2*+ fibroblasts, including producers of extracellular matrix components and a pro-inflammatory subset. Further, our data demonstrate that *SFRP2*+ fibroblasts are a source of proteases such as cathepsin S encoded by *CTSS* that have been

**Fig. 7 | Ligand–receptor interaction analysis between the four main cell types in psoriatic skin. a** Heatmap showing the number of ligand–receptor pairs with a higher score in PP compared to NS among the cell types. The ligands were expressed by the cell types in the row, and the receptors were expressed by the cell types in the column. The color scale represents the number of ligand–receptor pairs. **b** Scatter pie plot showing the relative adjacency of the cell types identified in spatial-seq. **c** Heatmap showing the number of neighbors for each pair of cell types. **d** Circos plot showing the cytokine and growth factor ligand–receptor interactions with a higher score in PP compared to NS. The ligands are expressed by the keratinocytes, the receptors are expressed by the other cell types. **e** Circos plot showing the cytokine and growth factor ligand–receptor interactions with a higher score in PP compared to NS. The ligands are expressed by the fibroblasts, the receptors are expressed by the other cell types. **f** Circos plot showing the cytokine and growth factor ligand–receptor interactions with a higher score in PP compared to NS. The ligands are expressed by the myeloid cells, the receptors are expressed by the other cell types. **g** Circos plot showing the cytokine and growth factor ligand–receptor interactions with a higher score in PP compared to NS. The ligands are expressed by the T cells, the receptors are expressed by the other cell types.

demonstrated to have a role in activating IL-36G[56]. Consistent with such a role, fibroblast expression of these pro-inflammatory mediators was most prominently located in the tips of the dermal papillae, an area of psoriatic skin that has been shown to have the highest number of inflammatory cells[57] and is in close proximity to the overlying epidermis. The *SFRP2*⁺ inflammatory fibroblasts in psoriasis appear analogous to an inflammatory fibroblast population in murine skin, with the latter expressing *Camp*[58], encoding an antimicrobial peptide that contributes to inflammation in psoriasis[59,60]. In addition, the *SFRP2*⁺ inflammatory fibroblasts in psoriasis share with the *COL6A5*⁺ *COL18A1*⁺ fibroblasts in atopic dermatitis the expression of *CCL19* which is capable of recruiting *CCR7*⁺ *LAMP3*⁺ cDC2A[61]. Yet only the psoriatic fibroblasts express *CXCL12* and *CXCL1*, which can contribute to the recruitment of *CXCR4*⁺ Tc17 cells and neutrophils, respectively.

The contribution of keratinocytes to psoriasis has been appreciated for a long time, and psoriasis was initially considered to be a primary keratinocyte disorder[62]. PP keratinocytes are strikingly different transcriptionally from NS and PN keratinocytes. These shifts are likely secondary to the direct effects of psoriasis-associated cytokines that we detected as upstream regulators of the keratinocyte pseudo-time trajectory, especially IFN-γ, IL-17A, and TNF[12]. These cytokines are postulated to "feed-forward" by activating keratinocytes to secrete additional pro-inflammatory cytokine and chemokine mediators[12]. Our previous work has demonstrated that the IL-17 receptor, a hetero-dimeric receptor composed of IL-17RA and IL-17RC, is most prominently expressed in the supraspinous layer of the psoriatic epidermis[13]. Consistent with this, we now demonstrate that the strongest IL-17A and TNF downstream responses in psoriasis are in the supraspinous compartment. Strikingly, IL-36 activity, and IL-36 to IL-36R ligand–receptor interactions in our single-cell and spatial-seq data, were also strictly compartmentalized within the spinous and supraspinous compartment of psoriatic skin. Using CRISPR-Cas9-induced KO of *IL36A*, *IL36G*, and *IL1RL2* (IL-36R), we demonstrate that IL-36R is essential for amplifying both IL-17A and TNF responses in keratinocytes. Interestingly, the deletion of IL-36A and IL-36G, two of the most prominent IL-36 members in psoriatic skin[7], had the opposite effect on TNF and IL-17A responses. Whereas deletion of IL-36G suppressed IL-17A and TNF responses, deletion of IL-36A led to the amplified response. IL-36 cytokines are secreted as full-length relatively inactive cytokines, but extracellular neutrophil-derived proteases can cleave IL-36 cytokines to their truncated forms markedly increasing their biologic activity[63]. Notably, IL-36G has been shown to be processed by the keratinocyte protease cathepsin S[56], setting up a potential interaction, or competition, between different IL-36 members under different protease conditions, where inactive full-length IL-36A may be competing against truncated, biologically active, isoforms such as cathepsin S processed IL-36G. Our data suggest that the PP fibroblasts express cathepsin S, further emphasizing the role of fibroblast–keratinocyte crosstalk in amplifying inflammatory responses in psoriatic skin.

In addition to the changes in the stromal and epithelial cell types, our data also provide unprecedented detail on the specific inflammatory cell populations involved, including both myeloid and T cells. Six populations of myeloid cells were identified: LC, cDC1, cDC2A, cDC2B, CD16⁺ DCs, and PVM. We observed a decreased proportion of LCs in

psoriatic skin, but they are still present in psoriatic plaques, especially at the dermal–epidermal junction around the tips of the dermal papillae, as shown in our IHC. Whether the lower LC proportion is due to an absolute reduction in LCs or their dilution from an increased proportion of other immune cell populations remains unclear[64]. PVM were slightly decreased in PP, but this is likely due to dilution, as there is an overall increase in the number of myeloid cells in PP skin. PVM were prominent on IHC, especially in the dermal papillae in a peri-vascular distribution. These cells have been shown to be a major source of TNF and IL-12/IL-23[65]. cDC2A myeloid cells and the CD16⁺ DC subset were the most expanded myeloid cell populations in psoriatic skin. The CD16⁺ DC subset was enriched for *CXCL9* and *CXCL10* expression, important chemokines for induction of Th1/Th17 immune responses[66]. By contrast, cDC2A cells were characterized by expression of *LAMP3*, *CD1B*, and *CCR7*, the co-stimulatory molecules *CD80* and *CD274* (PD-L1), as well as the pro-inflammatory cytokines *IL15* and *IL32*. *LAMP3* is a marker of mature dendritic cells and, along with co-stimulatory molecules, may play a role in expanding autoreactive T cells in psoriatic skin. IL-32 has been shown to be increased in psoriatic skin[67] and in inducing the expression of pro-inflammatory cytokines including TNF, IL-1, and IL-6[68].

These data demonstrate how scRNA-seq data can be integrated with spatial-seq transcriptomic information to define the localization of specific cell populations and inflammatory pathways in inflamed skin and within specific epidermal compartments. Our integrated scRNA-seq and spatial transcriptomics data, as expected, identified keratinocytes and keratinocyte responses within the epidermis, and immune and endothelial cells in the superficial dermis. It also accurately located smooth muscle and eccrine gland cell populations in the skin. The spatial data demonstrated co-localization of T cells and myeloid cells in the upper dermis and in close proximity to endothelial cells, consistent with the relative abundance of inflammatory cell infiltration in the superficial psoriatic dermis. The ligand–receptor interactions corresponded closely with the spatial localization of the cells, showing the highest number of ligand–receptor pairs between keratinocytes, fibroblasts, myeloid cells, and T cells. Most of the known keratinocyte-activating T cell-derived cytokines (reviewed by Hawkes et al.[12]) were captured in our ligand–receptor analysis, such as T cell-derived IL-17 effect on keratinocytes. In addition, our analysis also outlined the cytokines and chemokines produced by myeloid cells and fibroblasts and defined their interactions with particular immune cell and stromal cell subsets. Thus, the ligand–receptor interactions described here highlight a complex network that complements the major pathways that drive critical pathogenic mechanisms, such as abnormal epidermal differentiation, in psoriasis.

Our study has several limitations. First, the quality of the spatial-seq datasets is restricted by low detection rate, low resolution, incomplete coverage, and ambient RNA contamination. Thus, our analyses were limited to deconvolution, displaying highly expressed genes, and calculating module scores. Further analyses, including spatial proximity of cell types and co-localization of ligands and receptors, were prohibited by these drawbacks. Second, although the principle of Seq-Scope permits high resolution, we did not obtain high-quality datasets after multiple rounds of optimization, limiting the

analysis to module score calculation. Last, we did not detect neutrophils by scRNA-seq, rendering the inference of neutrophil interactions with the key fibroblast and keratinocyte subtypes unvalidated. These limitations notwithstanding, our datasets provide an unprecedented view of psoriasis pathology, identify, and characterize the main cellular players and define their interactions including a major role of fibroblast–keratinocyte crosstalk involving IL-17/IL-36 responses as an amplification mechanism in psoriasis, and demonstrate how different mechanisms drive different aspects of the disease process. Furthermore, through the integration of genetic and spatial information, these data provide a unique resource for future investigations into psoriasis pathogenesis.

## Methods

### Human samples and IRB statement
Fourteen patients with chronic plaque psoriasis and 11 healthy controls were recruited for this study. Patients were off systemic treatment and off any topical agents for at least 2 weeks prior to study time. The study was approved by the University of Michigan Institutional Review Board (IRB), and all patients provided written informed consent. 6-mm punch biopsies were taken from lesional and non-lesional skin. The study was conducted according to the Declaration of Helsinki Principles.

### Immunohistochemistry
Paraffin-embedded tissue sections from excisional biopsies from patients with psoriasis and healthy control skin were heated at 60 °C for 30 min, de-paraffinized, and rehydrated. Slides were placed in pH 6 antigen retrieval buffer and heated at 125 °C for 30 s in a pressure cooker water bath. After cooling, slides were treated with 3% $H_2O_2$ (5 min) and blocked using 10% goat serum (30 min), followed by an overnight incubation (4 °C) with the primary antibody of interest. Antibodies used include: CLEC9A (ThermoFisher Scientific, 55451-I-AP) (1:70), CLEC10A (ThermoFisher Scientific, TA810180) (1:150)(Clone:O-TI2B10), CD16 (Abcam, AB183354SP175)(2 μg/ml)(Clone: SP175), CD163 (ThermoFisher Scientific, MA5-11458)(1:25)(Clone:10D6), SFRP2 (Lifespan Biosciences, LS-C794043)(1:150) (Clone: OTI6E1), SFRP4 (Lifespan Biosciences, LS-C408100)(1 μg/ml) (Epitope: aa22-303), COL11A1 (ThermoFisher Scientific, PA5-68410)(2 μg/ml), Langerin (ThermoFisher Scientific, PA5-82422)(1:200), LAMP3 (ThermoFisher Scientific, PA5-84069)(1:50), FGFR3 (Abcam, AB231442)(2 μg/ml), FGFR2 (ThermoFisher Scientific, 13042-I-AP)(2 μg/ml), FGF2 (ThermoFisher Scientific, OSG00014W)(1:300), FGF7 (ThermoFisher Scientific, PA5-83670) (2 μg/ml). IL-17A (Lifespan Biosciences, LS-C104427)(5 μg/ml), CD4 (LifeSpan Biosciences, LS-C87801)(1:20) (Clone: 4B12), CD8 (Thermofisher scientific, MA5-13473)(1:50)(Clone: C8/144B), Goat IgG (Jackson ImmunoResearch Labs, 005-000-003), mouse IgG1 k (BioLegend, 400102)(Clone: MOPC-21). Slides were then washed and treated with appropriate secondary antibodies, peroxidase (30 min), and diaminobenzidine substrate, before imaging.

### Single-cell RNA sequencing library preparation, sequencing, and alignment
Generation of single-cell suspensions for single-cell RNA-sequencing (scRNA-seq) was performed as follows: NS tissue was obtained from healthy donors, and PP and PN samples were obtained from patients with psoriasis. Samples were incubated overnight in 0.4% dispase (Life Technologies) in Hank's Balanced Saline Solution (Gibco) at 4 °C. Epidermis and dermis were separated. Epidermis was digested in 0.25% Trypsin-EDTA (Gibco) with 10 U/mL DNase I (Thermo Scientific) for 1 h at 37 °C, quenched with FBS (Atlanta Biologicals), and strained through a 70 μM mesh. Dermis was minced, digested in 0.2% Collagenase II (Life Technologies) and 0.2% Collagenase V (Sigma) in plain medium for 1.5 h at 37 °C, and strained through a 70 μM mesh. Epidermal and dermal cells were combined in 1:1 ratio and libraries were constructed by the University of Michigan Advanced Genomics Core on the 10X

Chromium system with chemistry v2 and v3. Libraries were then sequenced on the Illumina NovaSeq 6000 sequencer to generate 150 bp paired-end reads. Data processing including quality control, read alignment (hg38), and gene quantification was conducted using the 10X Cell Ranger software. The samples were then merged into a single expression matrix using the cellranger aggr pipeline.

### Cell clustering and cell type annotation
The R package Seurat (v3.1.2)[19] was used to cluster the cells in the merged matrix. Cells with <500 transcripts or 100 genes, or more than 10% of mitochondrial expression were first filtered out as low-quality cells. The NormalizeData was used to normalize the expression level for each cell with default parameters. The FindVariableFeatures function was used to select variable genes with default parameters. The FindIntegrationAnchors and IntegrateData functions were used to integrate the samples prepared using different 10X Chromium chemistries. The ScaleData function was used to scale and center the counts in the dataset. Principal component analysis (PCA) was performed on the variable genes, and the first 20 PCs were used for cell clustering and uniform manifold approximation and projection (UMAP) dimensional reduction. The clusters were obtained using the FindNeighbors and FindClusters functions with the resolution set to 0.5. The cluster marker genes were found using the FindAllMarkers function. The cell types were annotated by overlapping the cluster markers with the canonical cell type signature genes. To calculate the disease composition based on cell type, the number of cells for each cell type from each disease condition was counted. The counts were then divided by the total number of cells for each disease condition and scaled to 100 percent for each cell type. Differential expression analysis between NS and PP or PN and PP was carried out using the FindMarkers function.

### Cell type sub-clustering
Sub-clustering was performed on the abundant cell types. The same functions described above were used to obtain the sub-clusters. Sub-clusters that were defined exclusively by mitochondrial gene expression, indicating low quality, were removed from further analysis. The subtypes were annotated by overlapping the marker genes for the sub-clusters with the canonical subtype signature genes. Ingenuity pathway analysis was applied to the differentially expressed genes to determine the canonical pathways and the potential upstream regulators. The upstream regulators with an activation $z$ score $\geq 2$ were considered significantly activated. The module scores were calculated using the AddModuleScore function on the genes induced by the intended cytokine from bulk RNA-seq analysis. The cytokine-induced genes were obtained from studies published by Billi et al.[69]. Specifically, keratinocytes were cultured and stimulated with the cytokine. Bulk RNA-seq and differential expression analysis (DESeq2) were performed between the cytokine-stimulated and the control keratinocytes. Genes with adjusted $p$-value < 0.05 that are also highly expressed in the cytokine-stimulated keratinocytes were considered the cytokine-induced genes. The cytokine-induced genes were listed in Supplementary Data 7.

### Pseudotime trajectory construction
Pseudotime trajectory was constructed using the R package Monocle[26]. The raw counts for cells were extracted from the Seurat analysis and normalized by the estimateSizeFactors and estimateDispersions functions with the default parameters. Genes with average expression larger than 0.5 and detected in more than 10 cells were retained for further analysis. Variable genes were determined by the differentialGeneTest function with a model against the subtype identities for the NS or PP keratinocytes. The orders of the cells were determined by the orderCells function, and the trajectory was constructed by the reduceDimension function with default parameters.

The heatmap showing specific genes along pseudotime was plotted using the plot_pseudotime_heatmap function. Differential expression between pseudo-time states was carried out using the Seurat function FindMarkers. Ingenuity Pathway Analysis was used to determine the upstream regulators for the DEGs.

## Psoriasis-associated genetic risk variants analysis

We acquired 65 single nucleotide polymorphism (SNP) loci detected by GWAS in psoriasis[4]. The eQTL datasets were downloaded from GTEx, and the Skin_Not_Sun_Exposed_Suprapubic.v7.signif_variant_gene_pairs.txt file was used to look for the genes significantly associated with the 65 SNPs. A total of 46 genes associated with these SNPs were identified. The nearest gene to SNP loci was also considered in the analysis, which yielded another 65 associated genes. The average expression of these genes was calculated by each cell type split by disease conditions. Genes with an average expression <0.01 in all cell types were filtered out as lowly expressed genes. After the filter, 27 eQTL-associated genes and 57 nearest genes were plotted.

## Ligand–receptor interaction analysis

CellphoneDB (v2.0.0)[22] was applied for ligand–receptor analysis. The Seurat normalized counts and sub-cluster annotation for each cell were input into cellphoneDB to determine the potential ligand–receptor pairs. Pairs with p-value > 0.05 were filtered out from the further analysis. To explore the specificity of ligand–receptor pairs in NS and PP, only the NS and PP cells were used. The sub-clusters were divided into NS-specific sub-cluster (with NS composition > 70%), PP-specific sub-cluster (with PP composition > 70%), and mixed sub-cluster. CellphoneDB was then run on the NS cells from the NS-specific sub-clusters and PP cells from the PP-specific sub-clusters. The pairs from the sub-clusters for the same cell type were merged to find ligand–receptor pairs between the major cell types. The pairs with higher interaction scores in PP were plotted. For the ligand–receptor analysis within the keratinocyte subtypes, the Seurat normalized counts and subtype annotation for each cell were used. CellphoneDB was run on all the NS keratinocyte subtypes and all the PP keratinocyte subtypes. The ligand-receptor pairs with higher interaction scores in PP were plotted.

## Spatial sequencing library preparation

Skin samples were frozen in OCT medium and stored at −80 °C until sectioning. Optimization of tissue permeabilization was performed on 20-μm-thick sections using Visium Spatial Tissue Optimization Reagents Kit (10X Genomics, Pleasanton, CA, USA), which established an optimal permeabilization time to be 24 min. Samples were mounted onto a Gene Expression slide (10X Genomics), fixed in ice-cold methanol, stained with hematoxylin and eosin, and scanned under a microscope (Keyence, Itasca, IL, USA). Tissue permeabilization was performed to release the poly-A mRNA for capture by the poly(dT) primers that are precoated on the slide and include an Illumina TruSeq Read, spatial barcode, and unique molecular identifier (UMI). Visium Spatial Gene Expression Reagent Kit (10X Genomics) was used for reverse transcription to produce spatially barcoded full-length cDNA and for second-strand synthesis followed by denaturation to allow a transfer of the cDNA from the slide into a tube for amplification and library construction. Visium spatial single-cell 3′ gene expression libraries consisting of Illumina paired-end sequences flanked with P5/P7 were constructed after enzymatic fragmentation, size selection, end repair, A-tailing, adaptor ligation, and PCR. Dual Index Kit TT Set A (10X Genomics) was used to add unique i7 and i5 sample indexes and generate TruSeq Read 1 for sequencing the spatial barcode and UMI and TruSeq Read 2 for sequencing the cDNA insert, respectively.

For Visium for FFPE, the skin samples were formalin-fixed and paraffin-embedded. 5-micron thick sections were mounted onto a Visium Gene Expression slide (10X Genomics), baked at 42 °C for 3 h, and dried in a desiccator at room temperature overnight. For deparaffinization, the slide was incubated at 60 °C for 2 h, immersed in xylene, and rehydrated in an ethanol gradient. H & E staining was then performed using Mayer's hematoxylin (Millipore Sigma), bluing reagent (Dako, Agilent), and alcoholic eosin (Millipore Sigma). Stained slides were scanned under a microscope, followed by decrosslinking using 0.1 N HCl and TE Buffer (pH 9.0) to release RNA that was sequestered by formalin. Human whole transcriptome probe panel (10x) that consisted of a pair of specific probes (5′ containing Small RNA Read 2S and 3′ containing poly-A) for each gene was hybridized to RNA. Probe pairs were then ligated to seal the junctions between them and to form the single-stranded ligation products. The samples were treated with RNase and permeabilized to release the ligation products. Poly-A portion of the products was then captured by the poly(dT) regions of the capture probes precoated on the Visium slide that also include an Illumina Read 1, spatial barcode, and unique molecular identifier (UMI). Probes were extended to produce spatially barcoded ligated probe products and released from the slide for indexing via Sample Index PCR and final library construction and sequencing. Visium Spatial Gene Expression FFPE libraries consisted of Illumina paired-end sequences flanked with P5/P7. The 16-bp Spatial Barcode and 12-bp UMI were encoded in Read 1, while Read 2S was used to sequence the ligated probe insert.

## Spatial sequencing data analysis

After sequencing, the reads were aligned to the human genome (hg38), and the expression matrix was extracted using the spaceranger pipeline. Seurat was then used to analyze the expression matrix. Specifically, the SCTransform function was used to scale the data and find variable genes with default parameters. PCA and UMAP were applied for dimensional reduction. The FindTransferAnchors function was used to find a set of anchors between the scRNA-seq data and spatial-seq data, which were then transferred from the scRNA-seq to the spatial-seq data using the TransferData function. The major cell types obtained in the scRNA-seq data were used to annotate the spatial-seq data. The predicted cell type composition for each spot was then used to cluster the spots by the k-means algorithm. The clusters were annotated based on the average cell type prediction score across all the spots in the cluster. The cluster marker genes were found using the FindAllMarkers function. Differential expression analysis between the NS and PP keratinocytes was performed using the FindMarkers function. The common DEGs comparing NS to PP keratinocytes in both scRNA-seq and spatial-seq were plotted.

The current version of this technique is limited by low detection rate, low resolution, incomplete coverage, and ambient RNA contamination. Thus, the analyses were limited to deconvolution, displaying highly expressed genes, and calculating module scores. The ambient RNA contamination might induce explains the detection of some highly expressed transcripts in nearby spots, such as S100A7 in our datasets. In the second psoriasis spatial sample (PP2), there was a technical issue that the epidermis is folded, resulting in positive detection of epidermis signals in part of the dermis area.

## Generation of CRISPR KO lines in N/TERTs keratinocytes

N/TERTs[70], an immortalized keratinocytes line, was used with the kind permission of Dr. James G. Rheinwald for the generation of knock-out (KO) cell lines using non-homologous end joining via CRISPR/Cas9. N/TERTs were grown in Keratinocyte-SFM medium (ThermoFisher #17005-042) supplemented with 30 μg/ml bovine pituitary extract, 0.2 ng/ml epidermal growth factor, and 0.3 mM calcium chloride.

Single-guide RNA (sgRNA) target sequence was developed using a web interface for CRISPR design (https://portals.broadinstitute.org/gpp/public/analysis-tools/sgrna-design). The pSpCas9 (BB)−2A-GFP (PX458) was a gift from Feng Zhang (Addgene plasmid # 48138) and was used as a cloning backbone. We followed the CRISPR/Cas9

protocol described by us previously[71] to generate KO KCs. In short, sgRNA target sequences for *IL36A* (GGATATCAATCATCGGGTGT), *IL36G* (TATCACATGCAAGTATCCAG), and *IL1RL2* (AAATGTCCTTGC ATCCATCTA) were designed with Broad Institute Guide RNA design tools and sgRNA specific oligos were purchased from Millipore-Sigma. Those oligos were annealed and the annealed oligonucleotides were inserted into the cloning vector Px458 following the protocol described earlier2. Ligated plasmids were transformed into competent E. coli (ThermoFisher # C737303) and then plated on an LB-agar plate overnight. Multiple colonies were selected for plasmid preparation (Qiagen # 27106), and Sanger sequencing validated a plasmid with proper insertion of sgRNA target sequence. The plasmid with proper insertion was transfected into N/TERTs using the TransfeX transfection kit (ATCC # ACS4005). Single cells positive for GFP were sorted into 96-well plates using a MoFlo Astrios #1 cell sorter and grown up to ~50% confluency. Cells from 96-well plates were transferred into 12-well plates and grown to 50% confluency. DNA was extracted and PCR amplified using specific primers which cover the CRISPR target sequence. We selected KCs with homozygous mutation, which was validated by Sanger sequencing.

### Cytokine stimulation, RNA extraction, qRT-PCR
Control and KO keratinocytes were stimulated with IL-17A (R&D Systems # 317-ILB-050, concentration used−20 ng/ml), TNF (R&D Systems # 210-TA-020, concentration used−10 ng/ml) for 24 h. Cells were then harvested for RNA preparation. RNAs were isolated from cell cultures using Qiagen RNeasy plus kit (Cat # 74136). Reversed transcription was performed using a High-Capacity cDNA Transcription kit (ThermoFisher # 4368813). qPCR was performed on a 7900HT Fast Real-time PCR system (ThermoFisher) with TaqMan Universal PCR Master Mix (ThemoFisher # 4304437) using TaqMan primers (ThermoFisher Scientific, *DEFB4B*: Hs00823638_m1; *IL36G*: Hs00219742_m1; *IL36RN*: Hs01104220_g1; *S100A7*: Hs00161488_m1). *RPLPO* (ThermoFisher # Hs99999902_m1) was used as an endogenous control.

### Seq-Scope sample processing and analysis
A psoriasis skin biopsy sample was frozen in OCT medium and stored at −80 °C until sectioning. The current study utilized Seq-Scope[HISEQ], which is similar to the previously described version of Seq-Scope[MISEQ], but utilized the HISEQ2500 flow cells for Seq-Scope array generation, instead of the MISEQ flow cells. All the procedures are extensively described in the former paper[21], and the current study was performed almost identically; however, the following things were different between the two studies: (1) To perform the 1st-Seq process, HISEQ2500 sequencing was performed with 100 pM of HDMI32-oligo library, and generated 1.1 million clusters/mm$^2$ fully sequenced pixel density (PF clusters). (2) HISEQ2500 tiles have a rectangular shape and are arranged to form a minimally interrupted large imaging area. Therefore, images from individual tiles were assembled to present a larger image that can be aligned with the H&E histology results. (3) To accommodate the larger imaging area, the volume of the solutions was proportionally increased during the experiment. (4) During the secondary strand synthesis, we used an updated Randomer sequence (5′-TCAGACGTGTGCTCTTCCGATCTNNNNNNNNB-3′). Compared to the original Randomer (5′-TCAGACGTGTGCTCTTCCGATCTNNNNNNNN NN-3′), the updated Randomer sequence does not anneal to the poly-A region, enabling more efficient transcriptome alignment. Other experimental details and analysis methods are described in the previous study[21]. Step-by-step protocol and data processing tools of Seq-Scope[MISEQ] and Seq-Scope[HISEQ] are available at http://www.seq-scope.com.

The expression values were grouped into 10 µm grids, and each grid was considered a cell in the analysis. Grids with fewer than 30 genes detected were removed from the analysis. The NormalizeData from Seurat was used to normalize the expression level for each grid

with default parameters. Since most of the good-quality grids were in the epidermis, the analysis was limited to the epidermis region. The keratinocyte subtype module scores and the cytokine module scores were calculated using the AddModuleScore function. The correlation coefficient was calculated between the module scores.

### Statistics and reproducibility
Wilcoxon rank sum test (two-sided) was used in scRNA-seq marker gene analyses and differential expression analysis. Hypergeometric test was used for the enrichment analysis. The Benjamini–Hochberg procedure was used for false discovery rate (FDR) adjustment. For immunohistochemistry and immunofluorescence experiments, commercially available antibodies were used as outlined above using recommended conditions, and with appropriate isotype control (negative control).

### Reporting summary
Further information on research design is available in the Nature Portfolio Reporting Summary linked to this article.

## Data availability
The scRNA-seq data generated are available in GEO under accession number GSE173706. The Spatial-seq data generated are available in GEO under accession number GSE225475. All other data are available in the article and its Supplementary files or from the corresponding author upon request. Source data are provided with this paper.

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

## Acknowledgements

This work was supported by the National Psoriasis Foundation Psoriasis Prevention Initiative (NPF-PPI) Award (J.G., N.R.W., L.C.T., E.M., J.M.K.), Babcock Endowment Fund (L.C.T., J.T., J.E.G.), by the National Institute of Health: R01-AR069071 and R01-AR073196 (J.E.G.), P30-AR075043 (J.E.G.), K01-AR072129 (L.C.T.), R01-AR042742 and R01-AR054966 (J.T.E.), R01-AI022553, R01-ARO40312, R01-AR074302 and R01-AR074302 (R.L.M.), 5R01AR075959-02 (E.K.) and the A. Alfred Taubman Medical Research Institute (J.E.G., J.M.K.). A.C.B. is supported by the Dermatology Foundation. M.G. is supported by Swiss National Science Foundation (grant no. 310030B_182834). L.C.T. is supported by the Dermatology Foundation, Arthritis National Research Foundation, and National Psoriasis Foundation. M.K.S. is supported by the National Psoriasis Foundation. J.T.E. is supported by the Ann Arbor VA Healthcare System.

## Author contributions

Conceptualization—F.M., R.L.M., J.E.G.; Methodology—F.M., R.L.M., J.E.G., M.P., L.C.T., R.W.; Investigation—F.M., O.P., J.-E.H., A.C.B., X.X., R.W., M.G.-K., G.H., Y.J., P.W.H., J.X., J.H.L., M.K.S., Y.C., E.X., N.L.W., J.K.; Writing—F.M., M.P., R.L.M., J.E.G.; Resources—A.C.B., L.C.T., P.W.H., J.D.D., M.G., N.L.W., E.M., J.J.V., J.T.E., J.H.L., J.M.K., E.K., M.P., R.L.M., J.E.G.; Supervision—M.P. R.L.M., J.E.G.

## Competing interests

J.E.G. has received research support from Almirall, AnaptysBio, Novartis, AbbVie, Eli Lilly, Kyowa Kirin, Galderma, BMS, Janssen. J.E.G. has served as advisor to Pfizer, AbbVie, Eli Lilly, Galderma, Almirall, AnaptysBio, Novartis, Sanofi, BMS. J.M.K. has received research support from Janssen, BMS, and Q32 bio. J.M.K. has served as an advisor to Eli Lilly, BMS, AstraZeneca, GlaxoSmithKline, Ventus Therapeutics, Admirex Pharmaceuticals, Aurinia Pharmaceuticals, and Provention Bio. The remaining authors declare no competing interests.

## Additional information

[1]Division of Rheumatology, Department of Internal Medicine, University of Michigan, Ann Arbor, MI 48109, USA. [2]Department of Dermatology, University of Michigan, Ann Arbor, MI 48109, USA. [3]Department of Computational Medicine and Bioinformatics, University of Michigan, Ann Arbor, MI 48109, USA. [4]Department of Pathology, University of Michigan, Ann Arbor, MI 48109, USA. [5]Department of Molecular and Integrative Physiology, University of Michigan Medical School, Ann Arbor, MI 48109, USA. [6]Department of Dermatology, University Hospital of Lausanne, 1011 Lausanne, Switzerland. [7]Department of Dermatology, Case Western Reserve University, Cleveland, OH 44106, USA. [8]Department of Dermatology, University of California Davis, Sacramento,

CA, USA. ⁹Department of Pathology and Immunology, Washington University School of Medicine, St. Louis, MO 63110, USA. ¹⁰Ann Arbor Veterans Affairs Medical Center, Ann Arbor, MI 48105, USA. ¹¹Department of Molecular, Cell and Developmental Biology, University of California, Los Angeles, CA 90095, USA. ¹²Division of Dermatology, Department of Medicine, University of California, Los Angeles, CA 90095, USA. ¹³Department of Microbiology, Immunology and Molecular Genetics, University of California, Los Angeles, CA 90095, USA. ¹⁴These authors jointly supervised this work: Robert L. Modlin, Johann E. Gudjonsson. ✉e-mail: johanng@med.umich.edu

