## [Peer Review File · Nature Communications]

Single Cell and Spatial Sequencing Reveal Processes by which Keratinocytes and Fibroblasts Amplify Inflammatory Responses in PsoriasisEditorial Note: Parts of this Peer Review File have been redacted as indicated to maintain the confidentiality of figures generated during the review process.

REVIEWER COMMENTS

Reviewer #1 (Remarks to the Author):

Using scRNAseq and spatial transcriptomics, Gudjonsson and colleagues describe the cellular components of the inflammatory network in psoriasis - with a focus on fibroblasts, T cells, and keratinocytes - their interaction, and their localization within psoriatic skin.

The authors demonstrate that a subset of fibroblasts acquire an inflammatory phenotype in lesional psoriatic skin that allows them to recruit immune cells including T cells. These T cells, especially CD8 T cells, are shown to produce IL17A that directly stimulates keratinocytes to release IL36G. In turn, IL36G is able to amplify both IL17A- and TNF-induced responses leading to transcriptional changes in keratinocytes and subsequent hyperproliferation. This fibroblast-T cell-keratinocyte crosstalk occurs at the supraspinous layer of the epidermis at the tips of dermal papillae.

This study brings great amounts of new data, new insights, and draws conclusions on important new mechanisms underlying psoriasis pathogenesis. The study is certainly of major interest. The methodology is sound, the results support most of the author's conclusions, and the manuscript is well written.

However, in some parts of the the manuscript, it is difficult to follow the track of thought, particularly given the amount of data provided.

Critical points:

Whereas the IL36-induced program is clearly present in the psoriatic epidermis, the role of IL17A and TNF and the corresponding cytokine-producing T cells in the induction of this program is not convincing. In general, the T cell-keratinocytes crosstalk demonstration needs further corroboration.

The authors show positive correlation between the IL17A / IL36G modules and the supraspinous modules (Extended Data Fig. 4d), yet they need to show an absence of correlation between IL17A/IL36G and basal/spinous modules to demonstrate specificity for the supraspinous involvement of these pathways.

Although S100A7 seems to be detected in supraspinous keratinocytes spots in the first PP (Fig. 3g), it is mainly detected in the dermal compartment in the second and third PP (Extended Data Fig. 4e). Also, whereas S100A7, which is detected in supraspinous keratinocytes spots, is present all along the epidermis, IL36 is only detected in focal epidermal spots (close to folded epidermis for second PP) suggesting that the expression regulation of these two genes is very different. The authors claim that S100A7 and IL36G are representative IL-17- and TNF-induced genes, then they should perform spatial mapping of IL17A and TNF on these 3 PP sections to show their spatial relativity.

The direct link between IL17-producing T cells and the IL-17-signature induced in keratinocytes is not clear. First, according to the spatial location of T cells (Ext. Data Fig. 1e and Ext. Data Fig. 2), it seems that the expression of IL36G is detected in areas distant from T cells. Second, most of the T cells are located at the epidermal-dermal junction below the epidermis. So, one would expect that IL-17-induced genes are expressed by basal keratinocytes. The authors should bring additional analyses to clarify this point.

One finding made by the authors is that CD8 T cells are the major source of IL-17A in psoriasis (Fig. 5). However, data provided are not explicit enough. The authors should show the expression of CD4 in Fig. 5c and 5e to illustrate the distribution of CD4 vs CD8 T cells within the different clusters as well as the spatial mapping of both subsets to correlate with the IL17A and IL36G modules. Also, the expression of TNF should be shown in a similar way, as TNF is considered a key upstream regulator in this study.

To demonstrate the IL36G involvement in amplifying the IL17- and TNF-responses in keratinocytes, the authors use CRISPR/Cas9 to knock out for several IL36 family members in keratinocytes. Following CRISPR/Cas9, homozygous mutations were checked by sequencing but the authors should show efficient knockdown at the protein level (WB).

The fibroblasts in psoriatic skin are shown to transition from ECM-producing to cytokine-producing, the latter being located at the tips of the dermal papillae (Fig. 4). The authors should perform spatial mapping of these different fibroblast clusters to demonstrate the different location. They should also add IF images in Fig. 4i demonstrating the absence of cytokine production (cathepsin S, CCL19, CCL13, CXCL12) by dermal fibroblasts in contrast to the dermal papillae fibroblasts.

Throughout the manuscript, it is not clear, which cells mainly contribute to the production of type I IFNs, and where they are located. This should be at least discussed given the scoring through the epidermal layers that is inverted of the IL17 and IL36G modules.

Minor

Ext. Data Fig. 2a, and 2b, images swapped in the third panel

Seq-Scope results are very difficult to interpret (Ext. Data Fig. 4a, 4b, 4c). One misses the H&E image alone of the tissue used to understand the structure of the analyzed skin section.

Reviewer #2 (Remarks to the Author):

In this manuscript my Ma and colleagues, the authors have performed an exhaustive transcriptomic analysis of skin psoriatic lesions. They have combined scRNAseq along with some spatial transcriptomics and a variety of appropriate analytic tools. The authors should be lauded for the large number of samples obtained and the rigor of the analyses. Despite the obvious utility of these data to the community, I am hard pressed to find any highly impactful conclusions that move the field forward. Similar work in the past from the authors' own group and those of the Liao group have been reported. One new aspect the stratification of expression within the epidermis particularly as it related to IL17 and IL36. But this is not deeply explored. There is also concern that IL-17 producing CD4 T cells are absent from their analyses. These have been reported in the past and this discrepancy is not addressed.

Specific concerns:

- Could the authors please indicate clearly for all panels which datasets (i.e. spatial vs scRNAseq) are being analyzed.
- For Fig 1c,e: should be made clear if the genes shown are top DEGs or cluster-defining genes. If the latter, please also display the top 10-20 DEGs for each cluster (not just KCs)?
- Fig 1d: In the intro the authors discuss the importance of epidermal compartment in localization of autoinflammatory responses, yet their spatial transcriptomics data appear to exclude epidermal immune cells from analysis.
- How does their data in Fig 4 compare to fibroblast populations present in normal human skin as reported by Tabib et al JID 2018 or in AD from the guttman group? This group found SFRP2 to be expressed by numerous fibroblast subpopulations within normal skin. And what do the authors think is the significance of the fibroblast populations they define on the development of psoriasis given that fibrosis is not a prominent feature of this disease?
- Fig 5: the authors argue that CD8 T cells are the major source of IL-17A in psoriasis skin lesions, yet their T cell subcluster plot in Fig 5 panels a,e does not delineate CD4 T cells as a discernable population, nor does it define the "canonical T cell lineage markers" used for cluster 5, in which they detect the most IL-17A/F expression. It is not made clear if CD8 expression defines cluster 5 and if CD4+ T cells are excluded from cluster 5, and if so, in which cluster are they? Also, unclear if this is scRNA-seq or seq-scope data. If the latter, what spatial location(s) do the analyzed cells derive from? If epidermis, CD4 T cells may be underrepresented. This information should be clearly spelled out.

- The group of “stressed” T cells appears not fully characterized. What is the IL-17A/F in this population and how does it relate to the populations defined by the Clark lab?
- How does their clustering method differ from the CD8+ T cell scRNA-seq comparisons reported by Liu J et al (JACI 2021)?
- Also, the authors data does not agree with data previously reported by Kim et al JACI 2021 showing more il-17A production by CD4 t cells than CD8 t cells in PsO skin. Is this biologically relevant or is there a technical difference?

Reviewer #3 (Remarks to the Author):

Ma et al. performed scRNA-seq and spatial transcriptomics (ST), both Visium and Seq-scope, on a cohort of human patient psoriasis samples, including from non-lesional samples of these patients, as well as normal skin from healthy patients. Their analyses map out the landscape of cell types in these normal and diseased contexts, discovering distinct transcriptional signatures across epidermal keratinocyte (KC) compartments in psoriasis, identifying potential cytokines driving these programs through Ingenuity Pathway Analysis (IPA), including IFNA, IL17A, and IL36G. In vitro experiments demonstrate that TNF and IL17A KC responses are at least partially dependent on IL36G and its receptor IL1RL2 through CRISPR/Cas9 KO experiments. Spatial data supports IL17A- and IL36G-induced gene expression is higher in supraspinous KCs in psoriasis lesions. These are similar findings to those from Hughes et al., Immunity 2020, who also found that differentiated KCs in psoriasis exhibited stronger IL-17 responses, although this study offers additional mechanistic insights that IL36G and IL1RL2 are part of a feed forward pathway. The authors perform further analysis focused on myeloid cells, fibroblasts, and T cells in psoriasis, including cellular communication pathways operating among these cell types via ligand-receptor analyses and find that fibroblasts exhibit a pro-inflammatory state (also noted in the Hughes et al. 2020 paper). The authors conducted thoughtful analyses on these major cell types, particularly with respect to the scRNA-seq data, and experimental validation is focused on keratinocyte findings. Overall, the manuscript is well-presented. That said, the study also has notable limitations, including unclear benefit of the spatial data and unclear limits on spatial data depth and resolution. It would be helpful if authors included some discussion regarding these to guide the readers on how future studies could address and improve upon such limitations given the rapid expansion of this field. Since several major findings have been shown previously by the aforementioned Hughes et al. paper, the novelty of the study largely lies in the spatial dataset, but it does not appear these data are revealing of much additional insight. As the paper is presented as a data resource, some clarification regarding some of the data quality and analyses is additionally warranted. In its current state, its main strength serves as a scRNA-seq analysis paper that is consistent with previous studies, without major advances in spatial understanding of this disease, in my opinion. I discuss major and minor concerns below in more detail.

Major comments:

1. Although hyperproliferation of keratinocytes is a cardinal histological feature of psoriasis (as mentioned in the manuscript’s Introduction), I was surprised by the lack of any single-cell analyses on proliferating keratinocytes. Cell cycle genes are typically a strong source of heterogeneity across many scRNA-seq datasets, but the authors do not show or mention these at all. Were these cells removed from downstream analysis? Absent from the dataset? Spatially, it would be further interesting to know which compartment may exhibit hyperproliferation, and could the spatial data address this? I believe these are important questions to address.
2. There is an unproven assumption of any differentiation trajectory among the PP fibroblasts (clusters 3, 0, 6, and 2 in Figure 4) that authors focus on. For one, it is not clear that these clusters are enriched in psoriasis lesions and definitive quantification of the proportions of these cells across the NS, PN, and PP samples (i.e., what percentage of total NS/PN/PP fibroblasts does each cluster represent?) would be helpful to confirm enrichment rather than trying to gather a visual gestalt from the UMAP. Cluster 2 looks more mixed with other datasets, so it is possible that this cluster is not as psoriasis-specific and perhaps not a differentiation endpoint to focus on (I am speculating, but without

quantification, it's not clear). I am also wondering why authors chose the 3-0-6-2 trajectory when a trajectory from NS to PN to PP would seem like a more disease-relevant pathogenic trajectory (this was also the comparison used for the IPA predictions of cytokines regulating the SFRP2+ FB). Since all PP fibroblasts are already in a diseased state, it would be difficult to assign any temporal identity to them, so any assumption of a pseudotime trajectory is presumptuous (unless one could compare early vs. late clinical lesions). For example, cluster 3 fibroblasts are proposed to respond to the various predicted cytokines (Fig 4j) but seem to be enriched in psoriasis samples already. Why is it assumed that these give rise to cluster 6 fibroblasts? Why couldn't cluster 2 be the source of cluster 6 cells? Or cluster 1 (which seem increased in PN skin)? Similar validation of the effects of upstream cytokines on fibroblasts (which was done for KCs) could potentially help clarify this, although in situ validation would be more convincing given fibroblast plasticity in culture. Alternatively, clarifying spatial organization of these PP fibroblast subpopulations might also be a clue towards pathogenesis. The source of cluster 6 cells is undoubtedly an important and fascinating question, but I am having trouble buying all the assumptions involved in the pseudotime model proposed, and there is a lack of validation of the differentiation trajectory.

3. Along those lines, staining of the PP fibroblasts shown in Figure 4i (which focuses on cluster 6 fibroblasts) should be compared across the entire tissue (are cluster 6 fibroblasts specifically in upper dermis? Only the upper dermis is shown) as well as compared to PN and/or NS samples. The former comparison assesses the spatial specificity of these cells while the latter comparisons would provide additional evidence of any differentiation trajectory. In addition to this, given the presence of many other cell types in the upper dermis (myeloid cells, T cells, etc.), I am concerned that the use of vimentin as a co-stain is non-specific (vimentin is broadly expressed on these other cell types too). Some of these genes (CCL19/CTSS/CCL13) are also known to be expressed on myeloid cells. Based on images in Fig. 4, it is difficult to tell whether there are many co-stained cells (many cells appear single-positive), so quantification of double-positive cells would also be helpful (particularly compared to NS or PN skin). SFRP2 (or another more specific marker) co-staining with each of these genes with additional showing of merged signal and outlining the double-positive cells would be more convincing to demonstrate the positioning of these specific cells. According to IHC staining of SFRP2 in PP samples shown in Extended Data Fig. 7a, the majority of SFRP2 signal appears in different areas as shown in the IF from Fig. 4i (deeper in the dermis). Additional staining would make these data easier to reconcile.

4. Sequencing depth for the scRNA-seq and ST data are not reported. In addition, the spatial data overall is lacking metrics to help define its quality. While authors report average counts/spot and genes/spot on one sample, this info is missing for the others, which are also presented throughout. This is important to know as some spatial samples appear noisier than others with regarding to clustering (notably in Extended Fig. 2, keratinocyte clusters appear larger than the actual space occupied by epidermis) and S100A7 expression in Extended Fig. 4e looks diffuse in the dermis. This could indicate inaccuracies in the spatial data or technical issues and could lead to downstream analyses inaccuracies in deconvolution of spots. Reporting any limitations would further help the reader interpret these data and assess their value to the field.

5. Overall, the spatial data does not add much to the manuscript, i.e., what insights were gained from the spatial data that could not be gained from scRNA-seq alone? Epidermal compartments can be inferred from well-known transcriptional signatures of basal, spinous, or supraspinous KCs, and differentiating KCs exhibiting differential responses to IL-17 have been previously shown (Chiricozzi et al. 2014). It is also well known that there is an increase in immune cells in this disease compared to normal tissue; there is no mention of how spatial organization of various non-KC cell types may contribute to pathogenesis beyond their presence in the upper dermis. Certain claims in the discussion, such as referring to specific sets of cell types such as CCR7+ DCs and CXCR4+ CD8 T cells as "spatially proximate" to fibroblasts were not shown convincingly in any spatial data. Neutrophils are mentioned several times but are completely absent from both scRNA-seq and ST datasets or any targeted staining. Co-localization of ligands and receptor expression in the spatial data could further support these claims, however, these types of analyses are not undertaken. The authors should consider revising these claims or performing additional analyses and/or validation to demonstrate spatial proximity of these cell types. Given the higher resolution capabilities of Seq-scope, it seems like a missed opportunity to perform this technique on the more densely cellular areas of inflammation to more accurately deconvolve these cell types in spatial data and determine the

spatial relationships of these various cell types.

Minor concerns:

1. It would be helpful if the authors could more sharply discuss their findings in the context of recent scRNA-seq studies that also profiled psoriasis patient samples (Hughes et al, Immunity 2020, Reynolds et al. Science 2021, Gao et al. Cell Death and Disease 2021). Are your results confirmatory, do they extend previous findings, etc.?
2. How was the analysis done for determining the in vitro-derived KC gene signatures upon various cytokine treatment? There is no info on how these were derived in methods. Are there limitations when focusing on treating monolayer KCs as opposed to more differentiated KCs (as was found in Chiricozzi et al. 2014)?
3. Seq-scope analysis description and metrics is missing from Methods. Given this is a cutting-edge technology with far higher resolution than Visium, it would be very useful for the audience to know how many transcripts and genes per 10um area this method was capable of capturing, trade-offs in using this technique vs Visium, etc. These are important questions for the field of spatial genomics.
4. How were the capture spots from Seq-scope filtered for downstream analysis? How many spots are there? It's also not revealed which spots are included for correlation analysis in Extended Data Fig. 4d. These are all important to report.

Reviewer 1 (Remarks to the Author):

Using scRNAseq and spatial transcriptomics, Gudjonsson and colleagues describe the cellular components of the inflammatory network in psoriasis - with a focus on fibroblasts, T cells, and keratinocytes - their interaction, and their localization within psoriatic skin.

The authors demonstrate that a subset of fibroblast acquire an inflammatory phenotype in lesional psoriatic skin that allows them to recruit immune cells including T cells.

These T cells, especially CD8 T cells, are shown to produce IL17A that directly stimulates keratinocytes to release IL36G. In turn, IL36G is able to amplify both IL17A- and TNF-induced responses leading to transcriptional changes in keratinocytes and subsequent hyperproliferation. This fibroblast-T cell-keratinocyte crosstalk occurs at the supraspinous layer of the epidermis at the tips of dermal papillae.

This study brings great amounts of new data, new insights, and draws conclusions on important new mechanisms underlying psoriasis pathogenesis. The study is certainly of major interest. The methodology is sound, the results support most of the author's conclusions, and the manuscript is well written.

However, in some parts of the the manuscript, it is difficult to follow the track of thought, particularly given the amount of data provided.

Critical points:

1. Whereas the IL36-induced program is clearly present in the psoriatic epidermis, the role of IL17A and TNF and the corresponding cytokine-producing T cells in the induction of this program is not convincing. In general, the T cell-keratinocytes crosstalk demonstration needs further corroboration.

We thank the reviewer for this suggestion. We plotted the T cell marker genes, *CD3D*, *CD4*, *CD8A*, plus *IL17A* and *TNF*, as well as their receptors (*IL17RA* and *TNFRSF1A*) (**Figure R1**). The T cell marker genes were generally expressed at the dermal-epidermal junction in both psoriasis and normal skin. *IL17A* and *TNF* was also detected at the epidermal-dermal junction in psoriasis lesions but not in the normal skin lesions. We also detected the receptor genes for IL-17 (*IL17RA*) and TNF (*TNFRSF1A*), and detected their expression in the epidermis of the PP samples but not in the NS samples. These data suggest T cell production of IL-17 and TNF in psoriasis lesions, with the potential to trigger downstream responses.

In addition to the expression levels of the T cell marker genes, we also plotted the cytokine response module scores for IL-17A, IL-36G, and TNF in the spatial-seq samples and revealed their specific location in the epidermis (**Figure R3** in comment 3 from reviewer 1), which corroborated the T cell-keratinocyte crosstalk in psoriatic skin.

IL-17 is well known to induce expression of IL-36 cytokines (and synergize with them) in a 3D epidermal model (see PMID:29142248), and our CRISPR-Cas9 data using IL36R KO (*IL1RL2*) and IL-36G KO data shows the role of IL-36 cytokines in amplifying both IL-17 and TNF responses (See Figure 2F).

We added Figure R1 to Fig. 5e and revised the manuscript accordingly in the T cell results section.

[Editorial note: Figure R1 redacted]

2. The authors show positive correlation between the IL17A / IL36G modules and the supraspinous modules (Extended Data Fig. 4d), yet they need to show an absence of correlation between IL17A/IL36G and basal/spinous modules to demonstrate specificity for the supraspinous involvement of these pathways.

We thank the reviewer for this suggestion. We added the correlation between the IL-17A and IL-36G module scores with the basal, spinous and supraspinous module scores (**Figure R2**). The results demonstrate that there is an increased correlation between the IL-17A and IL-36G downstream responses moving from basal to spinous to supraspinous layers in psoriatic epidermis.

We added Figure R2 to Extended Data Fig. 4e and revised the manuscript in the keratinocyte results section accordingly.

[Editorial note: Figure R2 redacted]

3. Although S100A7 seems to be detected in supraspinous keratinocytes spots in the first PP (Fig. 3g), it is mainly detected in the dermal compartment in the second and third PP (Extended Data Fig. 4e). Also, whereas S100A7, which is detected in supraspinous keratinocytes spots, is present all along the epidermis, IL36 is only detected in focal epidermal spots (close to folded epidermis for second PP) suggesting that the expression regulation of these two genes is very different. The authors claim that S100A7 and IL36G are representative IL-17- and TNF-induced genes, then they should perform spatial mapping of IL17A and TNF on these 3 PP sections to show their spatial relativity.

We thank the reviewer for this suggestion. To illustrate the relationship between *IL36G* or *S100A7* expression and IL-17A or TNF responses, we plotted the IL-17A, IL-36G, and TNF module scores on the spatial-seq samples (**Figure R3**) and showed that the responses induced by these cytokines were strongest in the supraspinous layer, which is consistent to the expression patterns of *IL36G* and *S100A7*.

In terms of the detection of *S100A7* in the dermal spots of the spatial-seq samples, we observed that there is often ambient RNA contamination in spatial-seq experiments with the current Visium platform, which explains the detection of some highly expressed

transcripts in nearby spots, such as *S100A7* here. The detection of epidermal genes in dermal area could also be due to RNA spreading when the tissue was cut from epidermis to dermis. In terms of the second psoriasis spatial sample (PP2), there was a technical issue that the epidermis is folded, resulting in positive detection of epidermis signals in part of the dermis area.

We found that *S100A7* raw count number ranged from single digits to thousands in the spatial-seq spots in the PP samples, while it only reached up to 12 in NS1 and 2 in NS2. The highest raw counts were detected in the supraspinous keratinocyte spots in the PP samples (as shown by the “*S100A7* counts” column in **Figure R3**). In the spatial-seq analysis, we applied the SCTransform function from Seurat to normalize the raw counts by regularized negative binomial regression. This normalization method tends to correct extremely high values to match the intermediate ones, which could also account for the high normalized *S100A7* expression in the dermis of PP2 and PP3. To support these results, we plotted *S100A7* in the keratinocyte subtypes in scRNA-seq, which demonstrated its highest expression in the PP supraspinous layer (**Figure R4**).

We added Figure R3 to Fig. 3f and Extended Data Fig. 5b. Figure R4 was added to Extended Data Fig. 5c. We also clarified the technical issues in the methods section.

[Editorial note: Figure R3 redacted]

[Editorial note: Figure R4 redacted]

4. The direct link between IL17-producing T cells and the IL-17-signature induced in keratinocytes is not clear. First, according to the spatial location of T cells (Ext. Data Fig. 1e and Ext. Data Fig. 2), it seems that the expression of *IL36G* is detected in areas distant from T cells. Second, most of the T cells are located at the epidermal-dermal junction below the epidermis. So, one would expect that IL-17-induced genes are expressed by basal keratinocytes. The authors should bring additional analyses to clarify this point.

We thank the reviewer for this suggestion. The detection rate of the current Visium technique is much lower compared to the scRNA-seq technique, rendering the lowly expressed genes, such as *IL17A* and *IL17F*, even harder to be picked up. In the deconvolution analysis, the epidermis was overwhelmed by keratinocyte genes, i.e., the keratin genes. Therefore, the spots in the epidermis were assigned almost solely to keratinocytes although a small set of other cell type marker genes can be detected at a low level in some of the epidermis spots.

To address this issue, we plotted the T cell marker genes plus *IL17A* and *TNF* (**Figure R1**). Most of the T cell marker genes were expressed at the dermal-epidermal junction. Nevertheless, there were some *IL17A*-expressing and *TNF*-expressing T cells in the epidermis, suggesting a direct link between IL17-producing T cells and the IL17-induced responses in the epidermis. We also plotted the receptor genes for IL-17 (*IL17RA*) and TNF (*TNFRSF1A*), and demonstrate their abundant expression in the epidermis of the PP samples but not in the NS samples.

We added Figure R1 to Fig. 5e and revised the manuscript accordingly in the T cell results section.

5. One finding made by the authors is that CD8 T cells are the major source of IL-17A in psoriasis (Fig. 5). However, data provided are not explicit enough. The authors should show the expression of CD4 in Fig. 5c and 5e to illustrate the distribution of CD4 vs CD8 T cells within the different clusters as well as the spatial mapping of both subsets to correlate with the IL17A and IL36G modules. Also, the expression of TNF should be shown in a similar way, as TNF is considered a key upstream regulator in this study.

We thank the reviewer for this suggestion. We plotted the expression of *CD8A*, *CD4*, *IL17A*, and *TNF* in the T cells from the scRNA-seq data (**Figure R5**). By counting the cell numbers, we detected 85 cells that were *CD8A⁺IL17A⁺* and 9 cells that were *CD4⁺IL17A⁺*, confirming that IL-17 was mainly expressed by CD8 T cells in our dataset. TNF was expressed by several T cell subtypes, including part of the IL17-expressing T cells. We mapped the expression of these genes in the spatial-seq samples as shown in Figure R1 but could not determine the CD4/CD8 compositions of the IL17-expressing T cells due to the low detection rate.

We added Figure R5 to Extended Data Fig. 11c and revised the manuscript accordingly in the T cell results section.

[Editorial note: Figure R5 redacted]

6. To demonstrate the IL36G involvement in amplifying the IL17- and TNF-responses in keratinocytes, the authors use CRISPR/Cas9 to knock out for several IL36 family members in keratinocytes. Following CRISPR/Cas9, homozygous mutations were checked by sequencing but the authors should show efficient knockdown at the protein level (WB).

We thank the reviewer for this suggestion. We added the western blot results to support the knockdown at the protein level of *IL36G* and *IL1RL2* (encoding one protein of the IL36R) (**Figure R6**). We were unable to locate an antibody that detected IL-36A on a

Western Blot for IL-36A despite having tried three different commercially available IL-36A antibodies. Based on our Sanger Sequencing confirmation of the *IL36A* KO line, the mutation should prevent protein expression of an intact IL-36A protein, and we would therefore like to keep the *IL36A* KO data in the current figure, but we'd be happy to remove the *IL36A* data if the reviewer thinks that is necessary.

We added Figure R6 to Extended Data Fig. 3g and revised the manuscript accordingly in the keratinocyte results section.

[Editorial note: Figure R6 redacted]

7. The fibroblasts in psoriatic skin are shown to transition from ECM-producing to cytokine-producing, the latter being located at the tips of the dermal papillae (Fig. 4). The authors should perform spatial mapping of these different fibroblast clusters to demonstrate the different location. They should also add IF images in Fig. 4i demonstrating the absence of cytokine production (cathepsin S, CCL19, CCL13, CXCL12) by dermal fibroblasts in contrast to the dermal papillae fibroblasts.

We thank the reviewer for this suggestion. We plotted these genes in the PP spatial-seq samples (**Figure R7**). *SFRP2* and *CXCL12* were expressed by most of the PP fibroblasts, which is consistent with the scRNA-seq data. *CCL13*, *CCL19*, and *CTSS* were primarily detected close to the epidermis, consistent to the IF images that they were expressed at the tips of the dermal papillae. We also performed additional IF imaging to support this finding (**Figure R8**).

We added Figure R7 and Figure R8 to Extended Data Fig. 9 and revised the manuscript accordingly in the fibroblast results section.

[Editorial note: Figure R7 redacted]

[Editorial note: Figure R8 redacted]

8. Throughout the manuscript, it is not clear, which cells mainly contribute to the production of type I IFNs, and where they are located. This should be at least discussed given the scoring through the epidermal layers that is inverted of the IL17 and IL36G modules.

We thank the reviewer for this suggestion. It is generally known that the Type I IFN genes are difficult to be detected by scRNA-seq. We checked the expression of *IFNA2*, *IFNB1*, *IFNB2*, *IFNL1*, *IFNL2*, *IFNL3*, and *IFNK* in our scRNA-seq dataset, and found

only few cells that expressed these genes and at low expression levels. For example, *IFNA2* and *IFNB1* were detected in fewer than 10 cells out of a total of 67,378 cells (**Figure R9**).

[Editorial note: Figure R9 redacted]

Minor

9. Ext. Data Fig. 2a, and 2b, images swapped in the third panel

We thank the reviewer for this comment. We have corrected the figure.

10. Seq-Scope results are very difficult to interpret (Ext. Data Fig. 4a, 4b, 4c). One misses the H&E image alone of the tissue used to understand the structure of the analyzed skin section.

We thank the reviewer for this comment. We added the H&E staining to the figure.

Reviewer 2 (Remarks to the Author):

In this manuscript Ma and colleagues, the authors have performed an exhaustive transcriptomic analysis of skin psoriatic lesions. They have combined scRNAseq along with some spatial transcriptomics and a variety of appropriate analytic tools. The authors should be lauded for the large number of samples obtained and the rigor of the analyses. Despite the obvious utility of these data to the community, I am hard pressed to find any highly impactful conclusions that more the field forward. Similar work in the past from the authors' own group and those of the Liao group have been reported. One new aspect the stratification of expression within the epidermis particularly as it related to IL17 and IL36. But this is not deeply explored. There is also concern that IL-17 producing CD4 T cells are absent from their analyses. These have been reported in the past and this discrepancy is not addressed.

We thank the reviewer for this comment. We explored more on IL17 / IL36 gene expression and cytokine response using the spatial-seq data (**Figure R3**). The results illustrate strong stratification of IL17 and IL36 responses in the epidermis, specifically the supraspinous layer, which corroborates the findings from the scRNA-seq data. We further demonstrate using KO lines that IL-36G and the IL-36R are responsible for amplifying responses downstream of TNF and IL-17A. In addition, we characterized and expanded the role and function of fibroblasts in psoriasis pathogenesis. These results serve as novel findings that will help advance our understanding of psoriasis. In terms of whether CD4 or CD8 T cells that produce IL17, we address the question in the reviewer's comment 5.

Specific concerns:

1. Could the authors please indicate clearly for all panels which datasets (i.e. spatial vs scRNAseq) are being analyzed.

We thank the reviewer for this suggestion and have made it more clear in the figure legends.

2. For Fig 1c,e: should be made clear if the genes shown are top DEGs or cluster-defining genes. If the latter, please also display the top 10-20 DEGs for each cluster (not just KCs)?

We thank the reviewer for this comment. We have made it more clear in the legends that these are the cell type defining genes. We also added the DEGs between PP vs NS in each cell type in Supplementary Table 3.

3. Fig 1d: In the intro the authors discuss the importance of epidermal compartment in localization of autoinflammatory responses, yet their spatial transcriptomics data appear to exclude epidermal immune cells from analysis.

We thank the reviewer for this comment. The detection rate of the current Visium technique is much lower compared to the scRNA-seq technique, rendering the low expression genes, such as *IL17A* and *IL17F*, difficult to detect. In the deconvolution analysis of the epidermis, keratinocyte genes, i.e., the keratin genes were dominant. Therefore, the spots in the epidermis were assigned almost solely to keratinocytes although a small set of other cell type marker genes can be detected at a low level in some of the epidermis spots. To illustrate this limitation, we plotted the T cell marker genes plus *IL17A* and *TNF*. Most of the T cell marker genes were expressed at the dermal-epidermal junction. While there were some *IL17A*-expressing and *TNF*-expressing T cells in the epidermis, the signal is too weak for detailed analysis (**Figure R1**).

We added Figure R1 to Fig. 5e and revised the manuscript accordingly in the T cell results section.

4. How does their data in Fig 4 compare to fibroblast populations present in normal human skin as reported by Tabib et al JID 2018 or in AD from the guttman group? This group found *SFRP2* to be expressed by numerous fibroblast subpopulations within normal skin. And what do the authors think is the significance of the fibroblast populations they define on the development of psoriasis given that fibrosis is not a prominent feature of this disease?

We thank the reviewer for this comment. We adopted the annotation from the Tabib *et al.* paper on the fibroblast sub-clusters. In their study, *SFRP2/DPP4+* FB and *FMO1/LSP1+* FB were the most abundant fibroblast subpopulations. Our results were consistent with theirs and showed a differentiation from *SFRP2/DPP4+* FB to *FMO1/LSP1+* FB, as illustrated by Figure 4g. We believe that *SFRP2+* FB is serving more a role in extracellular matrix (ECM) deposition, which is not necessarily related to fibrosis. We hypothesize that the ECM-producing *SFRP2+* FB can differentiate into pro-inflammatory fibroblasts in psoriasis. Based on this hypothesis, we used pseudotime analysis to model the differentiation process and identified the potential drivers for this transition (Figure 4j). In the atopic dermatitis (AD) study from the Guttman group, they identified an inflammatory fibroblast subpopulation in AD lesions, which had a similar expression pattern as our fibroblast sub-cluster 6, marked with high expression of pro-inflammatory cytokines such as *CCL2* and *CCL19*.

We added these two studies as reference 27 and 61, respectively. We revised the manuscript accordingly in the discussion section.

5. Fig 5: the authors argue that CD8 T cells are the major source of IL-17A in psoriasis skin lesions, yet their T cell subcluster plot in Fig 5 panels a,e does not delineate CD4 T cells as a discernable population, nor does it define the “canonical T cell lineage markers” used for cluster 5, in which they detect the most IL-17A/F expression. It is not made clear if CD8 expression defines cluster 5 and if CD4+ T cells are excluded from cluster 5, and if so, in which cluster are they? Also, unclear if this is scRNA-seq or seq-scope data. If the latter, what spatial location(s) do the analyzed cells derive from? If epidermis, CD4 T cells may be underrepresented. This information should be clearly spelled out.

We thank the reviewer for this comment. Fig. 5a-d were generated based on the single cell RNA-seq data. To make the interpretation clearer, we added the UMAP showing the expression of *CD8A*, *CD4*, *IL17A*, and *IL17F* (**Figure R5**). These results demonstrate that IL-17 was mainly expressed by cells in cluster 5, and these cells were mostly CD8 positive but not CD4 positive. By counting the cell numbers, we detected 85 cells that were *CD8A⁺IL17A⁺* and 9 cells that were *CD4⁺IL17A⁺*, confirming that IL-17 was mainly expressed by CD8 T cells. As shown by the expression of *IL17A* in the spatial-seq samples (**Figure R1**), we could only detect *IL17A* in less than 10 spots in each sample. Thus, due to the low detection rate in the spatial-seq samples, it is impractical to further deconvolute the T cell spots by the T cell subtypes.

We added Figure R5 to Extended Data Fig. 11c, and we revised the manuscript accordingly in the T cell results section.

6. The group of “stressed” T cells appears not fully characterized. What is the IL-17A/F in this population and how does it relate to the populations defined by the Clark lab?

We thank the reviewer for this comment. These cells expressed high levels of heat shock proteins (Figure 5c), suggesting they are under stress. As shown by Figure R5, *IL17A* and *IL17F* were only expressed by the Tc17 cells (cluster 5) but not in the “stressed” T cells. The study from Clark lab utilized high-throughput screening of the T cell receptor to determine the clonality of the IL-17A producing T cells in both resolved and active psoriatic lesions. This study is not a single cell study, and the cells were not sorted into CD4 and CD8 populations. Thus, we cannot directly compare our cells to theirs. Based on our newly added IF panels (**Figure R10**), we obtained consistent results that *IL17A*-expressing T cells mainly were located at the tips of dermal papillae.

We added Figure R10 to Extended Data Fig. 11d and revised the manuscript accordingly in the T cell results section.

[Editorial note: Figure R10 redacted]

7. How does their clustering method differ from the CD8+ T cell scRNA-seq comparisons reported by Liu J et al (JACI 2021)?

We used the same R package (Seurat) and ran the same process to cluster the cells. While the two analyses differ in some parameters, the same functions were applied. Thus, the clustering methods are almost identical between their study and ours.

8. Also, the authors data does not agree with data previously reported by Kim et al JACI 2021 showing more il-17A production by CD4 t cells than CD8 t cells in PsO skin. Is this biologically relevant or is there a technical difference?

We thank the reviewer for this comment. In their, Kim *et al.* harvested emigrating cells from human psoriasis skin after incubation in culture medium without enzyme digestion. This process could introduce bias to the CD4⁺ and CD8⁺ T cell compositions. Thus, we think that it is biologically relevant that our results are different from theirs.

Reviewer 3 (Remarks to the Author):

Ma et al. performed scRNA-seq and spatial transcriptomics (ST), both Visium and Seqscope, on a cohort of human patient psoriasis samples, including from non-lesional samples of these patients, as well as normal skin from healthy patients. Their analyses map out the landscape of cell types in these normal and diseased contexts, discovering distinct transcriptional signatures across epidermal keratinocyte (KC) compartments in psoriasis, identifying potential cytokines driving these programs through Ingenuity Pathway Analysis (IPA), including IFNA, IL17A, and IL36G. In vitro experiments demonstrate that TNF and IL17A KC responses are at least partially dependent on IL36G and its receptor IL1RL2 through CRISPR/Cas9 KO experiments. Spatial data supports IL17A- and IL36G-induced gene expression is higher in suprapapillary KCs in psoriasis lesions. These are similar findings to those from Hughes et al., Immunity 2020, who also found that differentiated KCs in psoriasis exhibited stronger IL-17 responses, although this study offers additional mechanistic insights that IL36G and IL1RL2 are part of a feed forward pathway. The authors perform further analysis focused on myeloid cells, fibroblasts, and T cells in psoriasis, including cellular communication pathways operating among these cell types via ligand-receptor analyses and find that fibroblasts exhibit a pro-inflammatory state (also noted in the Hughes et al. 2020 paper). The authors conducted thoughtful analyses on these major cell types, particularly with respect to the scRNA-seq data, and experimental validation is focused on keratinocyte findings. Overall, the manuscript is well-presented. That said, the study also has notable limitations, including unclear benefit of the spatial data and unclear limits on spatial data depth and resolution. It would be helpful if authors included some discussion regarding these to guide the readers on how future studies could address and improve upon such limitations given the rapid expansion of this field. Since several major findings have been shown previously by the aforementioned Hughes et al. paper, the novelty of the study largely lies in the spatial dataset, but it does not appear these data are revealing of much additional insight. As the paper is presented as a data resource, some clarification regarding some of the data quality and analyses is additionally warranted. In its current state, its main strength serves as a scRNA-seq analysis paper that is consistent with previous studies, without major advances in spatial understanding of this disease, in my opinion. I discuss major and minor concerns below in more detail.

Major comments:

1. Although hyperproliferation of keratinocytes is a cardinal histological feature of psoriasis (as mentioned in the manuscript's Introduction), I was surprised by the lack of any single-cell analyses on proliferating keratinocytes. Cell cycle genes are typically a strong source of heterogeneity across many scRNA-seq datasets, but the authors do not show or mention these at all. Were these cells removed from downstream analysis? Absent from the dataset? Spatially, it would be further interesting to know which compartment may exhibit hyperproliferation, and could the spatial data address this? I believe these are important questions to address.

We thank the reviewer for this suggestion. This is an excellent point that will improve our study. In the initial analysis, we excluded the proliferating cells because the cell

cycle effect adds another layer of variation to the data and makes the heterogeneity induced by different cell types or disease states hard to interpret. Now we added the proliferating keratinocytes back to the analysis and calculated the composition of this subtype across the disease states (**Figure R11**). The cycling keratinocytes were mainly derived from the PP samples that account for more than 50% of this subtype, supporting the notion that hyperproliferation of keratinocytes is an evident feature for psoriasis. We also plotted the cell cycle score in the spatial-seq samples (**Figure R11f**). Most of the cycling cells were in the basal layer in the epidermis for both PP and NS samples, and the cycling regions were thicker in the PP samples compared to the NS samples, suggesting stronger proliferation activity in psoriasis.

We added Figure R11 to Extended Data Fig. 7 and revised the manuscript accordingly in the keratinocyte results section.

[Editorial note: Figure R11 redacted]

2. There is an unproven assumption of any differentiation trajectory among the PP fibroblasts (clusters 3, 0, 6, and 2 in Figure 4) that authors focus on. For one, it is not clear that these clusters are enriched in psoriasis lesions and definitive quantification of the proportions of these cells across the NS, PN, and PP samples (i.e., what percentage of total NS/PN/PP fibroblasts does each cluster represent?) would be helpful to confirm enrichment rather than trying to gather a visual gestalt from the UMAP. Cluster 2 looks more mixed with other datasets, so it is possible that this cluster is not as psoriasis-specific and perhaps not a differentiation endpoint to focus on (I am speculating, but without quantification, it's not clear). I am also wondering why authors chose the 3-0-6-2 trajectory when a trajectory from NS to PN to PP would seem like a more disease-relevant pathogenic trajectory (this was also the comparison used for the IPA predictions of cytokines regulating the SFRP2+ FB). Since all PP fibroblasts are already in a diseased state, it would be difficult to assign any temporal identity to them, so any assumption of a pseudotime trajectory is presumptuous (unless one could compare early vs. late clinical lesions). For example, cluster 3 fibroblasts are proposed to respond to the various predicted cytokines (Fig 4j) but seem to be enriched in psoriasis samples already. Why is it assumed that these give rise to cluster 6 fibroblasts? Why couldn't cluster 2 be the source of cluster 6 cells? Or cluster 1 (which seem increased in PN skin)? Similar validation of the effects of upstream cytokines on fibroblasts (which was done for KCs) could potentially help clarify this, although in situ validation would be more convincing given fibroblast plasticity in culture. Alternatively, clarifying spatial organization of these PP fibroblast subpopulations might also be a clue towards pathogenesis. The source of cluster 6 cells is undoubtedly an important and fascinating question, but I am having trouble buying all the assumptions involved in the pseudotime model proposed, and there is a lack of validation of the differentiation trajectory.

We thank the reviewer for this comment. We added the disease compositions for each fibroblast sub-cluster (**Figure R12**), which demonstrates that sub-clusters 3, 0, 6, 2 were enriched in psoriasis, with more than 50% contributed from the PP samples.

We agree with the reviewer that generating a trajectory hypothesis in scRNA-seq is complicated and could be somewhat objective. We usually perform trajectory analysis based on two criteria: 1) the cells should form a continuum after dimensionality reduction, as shown by the UMAP in this study; 2) the hypothesis should make biological sense, for example, in the keratinocyte pseudotime analysis, we built the trajectories from basal to spinous to supraspinous layers for PP and NS cells, respectively. Here, we observed heterogeneous subsets within the *SFRP2*+ FB for the PP samples and tended to use pseudotime analysis to better model this continuous heterogeneity. Since psoriasis is not characterized by fibrosis, it makes biological sense that the fibroblasts went from a structural role (collagen-producing) to a pro-inflammatory role. Thus, we built the pseudotime trajectory along the sub-cluster 3-0-6-2 route.

We added Figure R12 to Extended Data Fig. 8d and revised the manuscript to clarify the rationale for the trajectory analysis in the fibroblast results section.

[Editorial note: Figure R12 redacted]

3. Along those lines, staining of the PP fibroblasts shown in Figure 4i (which focuses on cluster 6 fibroblasts) should be compared across the entire tissue (are cluster 6 fibroblasts specifically in upper dermis? Only the upper dermis is shown) as well as compared to PN and/or NS samples. The former comparison assesses the spatial specificity of these cells while the latter comparisons would provide additional evidence of any differentiation trajectory. In addition to this, given the presence of many other cell types in the upper dermis (myeloid cells, T cells, etc.), I am concerned that the use of vimentin as a co-stain is non-specific (vimentin is broadly expressed on these other cell types too). Some of these genes (*CCL19*/*CTSS*/*CCL13*) are also known to be expressed on myeloid cells. Based on images in Fig. 4, it is difficult to tell whether there are many co-stained cells (many cells appear single-positive), so quantification of double-positive cells would also be helpful (particularly compared to NS or PN skin). *SFRP2* (or another more specific marker) co-staining with each of these genes with additional showing of merged signal and outlining the double-positive cells would be more convincing to demonstrate the positioning of these specific cells. According to IHC staining of *SFRP2* in PP samples shown in Extended Data Fig. 7a, the majority of *SFRP2* signal appears in different areas as shown in the IF from Fig. 4i (deeper in the dermis). Additional staining would make these data easier to reconcile.

We thank the reviewer for this suggestion. We added more IF images co-staining these genes with *SFRP2* to support this finding (**Figure R8**). We also plotted these genes in the spatial-seq samples (**Figure R7**). *SFRP2* and *CXCL12* were expressed by most of the PP fibroblasts, which is consistent with the scRNA-seq data. *CCL13*, *CCL19*, and *CTSS* were primarily detected close to the epidermis, consistent to the IF images that they were expressed at the tips of the dermal papillae.

We added Figure R7 and Figure R8 to Extended Data Fig. 9 and revised the manuscript accordingly in the fibroblast results section.

4. Sequencing depth for the scRNA-seq and ST data are not reported. In addition, the spatial data overall is lacking metrics to help define its quality. While authors report average counts/spot and genes/spot on one sample, this info is missing for the others, which are also presented throughout. This is important to know as some spatial samples appear noisier than others with regarding to clustering (notably in Extended Fig. 2, keratinocyte clusters appear larger than the actual space occupied by epidermis) and S100A7 expression in Extended Fig. 4e looks diffuse in the dermis. This could indicate inaccuracies in the spatial data or technical issues and could lead to downstream analyses inaccuracies in deconvolution of spots. Reporting any limitations would further help the reader interpret these data and assess their value to the field.

We thank the reviewer for this suggestion. We added the quality metrics to Supplementary Table 2, which contains the number of cells/spots, average number of genes and UMIs detected per cell/spot, and the average mitochondrial expression percentage per cell.

In terms of the detection of S100A7 in the dermal spots of the spatial-seq samples, we observed that there is often ambient RNA contamination in spatial-seq experiments with the current Visium platform, which explains the detection of some highly expressed transcripts in nearby spots, such as S100A7 here. The detection of epidermal genes in dermal area could also be due to RNA spreading when the tissue was cut from epidermis to dermis. In terms of the second psoriasis spatial sample (PP2), there was a technical issue that the epidermis is folded, resulting in positive detection of epidermis signals in part of the dermis area.

We found that S100A7 raw count number ranged from single digits to thousands in the spatial-seq spots in the PP samples, while it only reached up to 12 in NS1 and 2 in NS2. The highest raw counts were detected in the supraspinous keratinocyte spots in the PP samples (as shown by the “S100A7 counts” column in **Figure R3**). In the spatial-seq analysis, we applied the SCTransform function from Seurat to normalize the raw counts by regularized negative binomial regression. This normalization method tends to correct extremely high values to match the intermediate ones, which could also account for the high normalized S100A7 expression in the dermis in PP2 and PP3. To support these results, we also plotted S100A7 in the keratinocyte subtypes in scRNA-seq, which demonstrated its highest expression in the PP supraspinous layer (**Figure R4**).

We added Figure R3 to Fig. 3f and Extended Data Fig. 5b. Figure R4 was added to Extended Data Fig. 5c. We also clarified the technical issues in the methods section

5. Overall, the spatial data does not add much to the manuscript, i.e., what insights were gained from the spatial data that could not be gained from scRNA-seq alone? Epidermal compartments can be inferred from well-known transcriptional signatures of basal, spinous, or supraspinous KCs, and differentiating KCs exhibiting differential responses to IL-17 have been previously shown (Chiricozzi et al. 2014). It is also well known that there is an increase in immune cells in this disease compared to normal tissue; there is no mention of how spatial organization of various non-KC cell types may contribute to pathogenesis beyond their presence in the upper dermis. Certain claims in the discussion, such as referring to specific sets of cell types such as CCR7+ DCs and CXCR4+ CD8 T cells as “spatially proximate” to fibroblasts were not shown convincingly in any spatial data. Neutrophils are mentioned several times but are completely absent from both scRNA-seq and ST datasets or any targeted staining. Co-localization of ligands and receptor expression in the spatial data could further support these claims, however, these types of analyses are not undertaken. The authors should consider revising these claims or performing additional analyses and/or validation to demonstrate spatial proximity of these cell types. Given the higher resolution capabilities of Seq-scope, it seems like a missed opportunity to perform this technique on the more densely cellular areas of inflammation to more accurately deconvolve these cell types in spatial data and determine the spatial relationships of these various cell types.

We thank the reviewer for this comment. The current Visium technique is limited by low detection rate, low resolution, incomplete coverage, and ambient RNA contamination. Thus, we are limited to some analyses like deconvolution, displaying highly expressed genes, and calculating module scores. To address these issues, we tried Visium spatial-seq on another PP sample with an FFPE tissue section, which was added to the study as spatial-seq sample PP4 and had higher coverage of gene expression compared to the other three frozen tissue sections (PP1-3). We plotted the cell type markers and ligand-receptor pairs in PP4 and confirmed the spatial proximity of these cell types and the pro-inflammatory cell-cell interactions induced by fibroblasts (**Figure R13**).

It is generally known that the standard 10X scRNA-seq pipeline could not pick up neutrophils (unless single nuclei were isolated for the library preparation). In our datasets, we did not detect neutrophils. Thus, we inferred the cell-cell interaction involving neutrophils and used dotted lines instead of solid lines to link the inferred neutrophils.

For Seq-scope, although the principle of this technique permits high resolution, we did not obtain high quality datasets after multiple rounds of optimization, hence restricting the analysis we could apply.

We add Figure R13 to Extended Data Fig. 14, and we clarified the technical limitations in the methods section for both Visium spatial-seq and Seq-Scope analysis. We also

[Editorial note: Figure R13 redacted]

Minor concerns:

6. It would be helpful if the authors could more sharply discuss their findings in the context of recent scRNA-seq studies that also profiled psoriasis patient samples (Hughes et al, Immunity 2020, Reynolds et al. Science 2021, Gao et al. Cell Death and Disease 2021). Are your results confirmatory, do they extend previous findings, etc.?

In the Hughes *et al.* and Reynolds *et al.* papers, they merged scRNA-seq datasets from multiple inflammatory skin diseases, and they focused on cell type annotations and the distinct features for each cell type in one disease compared to the others. Gao *et al.* studied psoriasis and described the cell types / subtypes in psoriatic skin samples. They emphasized the induced expression of major histocompatibility complex (MHC) genes, upregulated interferon (IFN), tumor necrosis factor (TNF) signaling in psoriatic skin to activate the neighboring dendritic cells. In our study, we confirmed most of the cell types identified by these studies and extended the mechanisms of IL-36 in augmenting IL-17/TNF responses in the supraspinous keratinocyte layer. Additionally, we highlighted the heterogeneity of fibroblasts and identified the potential drivers that could facilitate the differentiation of collagen-producing fibroblasts to cytokine-producing fibroblasts in psoriasis.

We have carefully revised our manuscript in terms of the confirmatory results and new findings in this study.

7. How was the analysis done for determining the in vitro-derived KC gene signatures upon various cytokine treatment? There is no info on how these were derived in methods. Are there limitations when focusing on treating monolayer KCs as opposed to more differentiated KCs (as was found in Chiricozzi et al. 2014)?

The signature lists were obtained from this paper “Nonlesional lupus skin contributes to inflammatory education of myeloid cells and primes for cutaneous inflammation”. Specifically, the keratinocytes were cultured and treated with different cytokines, and bulk RNA-seq was performed after cytokine treatment. The cytokine signatures were then obtained from differential expression analysis comparing the one cytokine treatment to the other cytokine treatments.

In terms of the monolayer KCs vs more differentiated KCs, we thank the reviewer for bringing up this intriguing question. We agree that the differentiation of the epidermis has impact on keratinocyte responses to certain cytokines. This includes responses to cytokines such as IL-17A, whose receptors are more highly expressed in the more differentiated layers (spinous/supraspinous) of the epidermis (PMID:33053333). The Chiricozzi 2014 paper also showed that reconstructed human epidermis has more robust responses to IL-17A (done by using microarray approaches and not bulk RNA-seq). We have extensive experience with these systems (PMID:33864770);

PMID:23190894; PMID:28259685; PMID:21242515) and have found 3D epidermal raft/organoid cultures have several limitations. They are hard to grow and there is considerable batch to batch variation. They have a low level "trauma" response that we have seen in our own experience with these systems (unpublished), and some of them (including the one in the Chiricozzi paper) consist of both keratinocytes and fibroblasts making a clear "keratinocyte-response" signature more difficult to obtain. In our experience, using bulk RNA-seq approaches, we do see that the IL-17 response had a decreased magnitude of responses in the monolayer KCs but was overall qualitatively similar to more differentiated KCs. These differences are less pronounced with other cytokine stimulations. Therefore, we do not think that using signatures from monolayer keratinocytes is an issue or one that will skew our results, but rather that it would possibly estimate the magnitude of the response (particularly for IL-17A) on the lower end of what it might be.

We added the details on cytokine signature scores to the methods section. The cytokine induced genes were listed in Supplementary Table 7.

8. Seq-scope analysis description and metrics is missing from Methods. Given this is a cutting-edge technology with far higher resolution than Visium, it would be very useful for the audience to know how many transcripts and genes per 10 μ m area this method was capable of capturing, trade-offs in using this technique vs Visium, etc. These are important questions for the field of spatial genomics.

We added the analysis description and the metrics for the Seq-scope. We detected an average of 148 genes and 181 transcripts in each Seq-scope grid. Although the principle of this technique permits high resolution, we did not obtain high quality datasets after multiple rounds of optimization, hence restricting the analysis we could apply.

We have clarified these limitations in the methods and discussion section.

9. How were the capture spots from Seq-scope filtered for downstream analysis? How many spots are there? It's also not revealed which spots are included for correlation analysis in Extended Data Fig. 4d. These are all important to report.

The expression values were grouped into 10 μ m grids, and each grid is considered a cell in the analysis. Grids with fewer than 30 genes detected were removed from the analysis. In total, we recovered 5,378 grids, and all of them were included for correlation analysis Extended Data Fig. 4d.

We added the quality metrics for the Seq-scope data in Supplemental Table 2. We also described the analysis in detail in the methods section.

REVIEWER COMMENTS

Reviewer #1 (Remarks to the Author):

The revised manuscript by Gudjonsson and colleagues is well written and presented. Using mainly scRNAseq and spatial transcriptomics, the authors describe the cellular components of the inflammatory network in psoriasis - with a focus on fibroblasts, T cells, and keratinocytes - their interaction, and their localization within psoriatic skin.

This study brings great amounts of new data (even more so, in the revised version), new insights, and draws conclusions on important new mechanisms underlying psoriasis pathogenesis. The methodology is sound, the results support most of the author's conclusions, and the manuscript is well written.

Most of the critical points have been addressed, and some important limitations have at least been discussed in the revised manuscript (though I am surprised that none of the IL-36A antibodies worked on Western Blot). The study is certainly of major interest and has been further improved by these revisions.

However, despite the amendments, a clear thread is still missing in parts of the manuscript and the track of thought remains sometimes difficult to follow, particularly given the amount of data provided. Some transitions between sections might help in that regard.

Reviewer #2 (Remarks to the Author):

The authors have responded relatively well to the first round of reviews. I still maintain that the authors have not adequately supported their claim that 'CD8+ Tc17 are the major source of IL17a in psoriatic skin'. CD4 T cells are omitted from their scRNA-seq clustering in fig 5a. This may result from differences between mRNA expression and protein as well as technique. This claim should be reduced and the topic needs to be discussed in greater detail. In addition, the comment from Reviewer 1 that "a direct link between IL-17producing T cells and the IL-17 signature induced in KCs not clear." has not been adequately addressed.

Reviewer #3 (Remarks to the Author):

I appreciate the efforts to address previous comments on the work and their willingness to acknowledge limitations of their work, particularly the spatial data. Unfortunately, the spatial data does not appear to be a major illuminating dataset. I also found that additional specific responses to previous comments fell short of assuaging my concerns, particularly the pseudotime analysis on fibroblasts and the additional IF staining shown.

1. Regarding pseudotime, I believe overall, there is needless speculation and the manuscript text makes strong assertions without enough evidence to back it up (particularly most of the text from lines 301-322). I would recommend at minimum revising the text prior to any potential publication. I find that the argument that psoriasis-enriched clusters should form the differentiation trajectory is still not well-defended. A keratinocyte trajectory can be constructed based on decades of prior work and knowledge. There is simply not enough knowledge about human fibroblast plasticity or origin of heterogeneity to make similar assumptions. The authors did not bother to entertain any other trajectories, such as one from NS to PN to PP, which to me makes far more biological sense than a model by which PP fibroblasts transition/differentiate within already inflamed psoriasis lesions (again, this is the comparison done for IPA analysis; so these two analyses are not consistent). Even within their current choice of trajectory, a trajectory from Cluster 2 (LSP1/FMO1+ fibroblasts present in normal skin) to Cluster 6 (inflammatory fibroblasts) is more plausible than a normal fibroblast being the terminal state. In my opinion, the pseudotime analyses should be scrapped entirely, or if included,

either the corresponding manuscript text needs to be re-written to reflect more uncertainty and the need for future studies, or additional validation/analysis is required. One potential way to make this data more convincing is to perform the same trajectory analyses on NS fibroblasts and show a lack of continuum or different results from PP. It seems a similar trajectory could be constructed with NS fibroblasts and show a similar result as PP fibroblasts, especially since most of these cells are present in NS, unless authors can prove this wrong.

2. Additional IF staining in Extended Fig. 9 appears to lack the same number of co-stained cells as those from Main Fig. 4 and no other areas besides the upper dermis to demonstrate specificity of the localization of these cells. This suggests either the SFRP2+ cells are in other areas of the dermis or there was a technical issue with the IF (e.g., poor antibody), and fails to support their assertion that there are abundant SFRP2+ fibroblasts in the upper dermis near epidermis in PP lesions that produce these chemokines and CTSS. As presented, the IF data suggest that other cell types in the upper dermis (such as myeloid cells) appear to be the dominant source of these chemokines/CTSS.

Reviewer #1 (Remarks to the Author):

The revised manuscript by Gudjonsson and colleagues is well written and presented. Using mainly scRNAseq and spatial transcriptomics, the authors describe the cellular components of the inflammatory network in psoriasis - with a focus on fibroblasts, T cells, and keratinocytes - their interaction, and their localization within psoriatic skin.

This study brings great amounts of new data (even more so, in the revised version), new insights, and draws conclusions on important new mechanisms underlying psoriasis pathogenesis. The methodology is sound, the results support most of the author's conclusions, and the manuscript is well written.

Most of the critical points have been addressed, and some important limitations have at least been discussed in the revised manuscript (though I am surprised that none of the IL-36A antibodies worked on Western Blot). The study is certainly of major interest and has been further improved by these revisions.

However, despite the amendments, a clear thread is still missing in parts of the manuscript and the track of thought remains sometimes difficult to follow, particularly given the amount of data provided. Some transitions between sections might help in that regard.

We thank the reviewer for these comments. We have carefully edited the manuscript to make it easier to follow and interpret.

Reviewer #2 (Remarks to the Author):

1. The authors have responded relatively well to the first round of reviews. I still maintain that the authors have not adequately supported their claim that 'CD8+ Tc17 are the major source of IL17a in psoriatic skin". CD4 T cells are omitted from their scRNA-seq clustering in fig 5a. This may result from differences between mRNA expression and protein as well as technique. This claim should be reduced and the topic needs to be discussed in greater detail.

We thank the reviewer for this comment. We are not trying to omit CD4 T cells in Figure 5a. In fact, as shown in **Figure 5a** and **Figure R1**, the majority of the cells in clusters 0, 1, and 2 were CD4 T cells, and these clusters were annotated as naïve T cells (cluster 0; *RGCC*, *CCR7*, *IL7R*) and stressed T cells (clusters 1 and 2; *DNAJB1*, *HSPA1A*, *HSPH1*). In addition, cluster 4 was annotated as Tregs (*FOXP3*, *TIGIT*, *BAFT*), which are also CD4 T cells. In terms of *IL17A*⁺ cells, there were a total of 94 cells, and 85 were *CD8A*⁺, only 9 were *CD4*⁺ (**Figure R2**). Based on these results as well as similar findings published by another group (Figure 1 in Liu *et al. J Allergy Clin Immunol*, 2021, PMID: 33309739), we believe that the scRNA-seq data reveals that CD8⁺ Tc17 are the major source of *IL17A* in psoriatic skin.

[Editorial note: Figure R1 redacted]

[Editorial note: Figure R2 redacted]

2. In addition, the comment from Reviewer 1 that “a direct link between IL-17-producing T cells and the IL-17 signature induced in KCs not clear.” has not been adequately addressed.

We thank the reviewer for this suggestion. We plotted the T cell marker genes, *CD3D*, *CD4*, *CD8A*, plus *IL17A* and its receptors *IL17RA* (**Figure R3**). The T cell marker genes were generally expressed at the dermal-epidermal junction in both psoriasis and normal skin. *IL17A* was also detected at the epidermal-dermal junction in psoriasis lesions but not in normal skin lesions. The receptor genes for IL-17 (*IL17RA*) were mainly detected in the epidermis of the PP samples but not in the NS samples. These data suggest T cell production of IL-17 induces the IL-17 signature in the epidermis (keratinocytes) in psoriasis skin.

Of note, this aspect was recently addressed in a paper published by Schabitz and colleagues in Nature Communications using spatial-sequencing data (PMID:36513651), and we believe our data aligns with the findings described there.

We have now cited this study (reference 36), added Figure R3 to Fig. 5e and revised the manuscript accordingly in the T cell results section.

[Editorial note: Figure R3 redacted]

Reviewer #3 (Remarks to the Author):

I appreciate the efforts to address previous comments on the work and their willingness to acknowledge limitations of their work, particularly the spatial data. Unfortunately, the spatial data does not appear to be a major illuminating dataset. I also found that additional specific responses to previous comments fell short of assuaging my concerns, particularly the pseudotime analysis on fibroblasts and the additional IF staining shown.

1. Regarding pseudotime, I believe overall, there is needless speculation and the manuscript text makes strong assertions without enough evidence to back it up (particularly most of the text from lines 301-322). I would recommend at minimum revising the text prior to any potential publication. I find that the argument that psoriasis-enriched clusters should form the differentiation trajectory is still not well-defended. A

keratinocyte trajectory can be constructed based on decades of prior work and knowledge. There is simply not enough knowledge about human fibroblast plasticity or origin of heterogeneity to make similar assumptions. The authors did not bother to entertain any other trajectories, such as one from NS to PN to PP, which to me makes far more biological sense than a model by which PP fibroblasts transition/differentiate within already inflamed psoriasis lesions (again, this is the comparison done for IPA analysis; so these two analyses are not consistent). Even within their current choice of trajectory, a trajectory from Cluster 2 (LSP1/FMO1+ fibroblasts present in normal skin) to Cluster 6 (inflammatory fibroblasts) is more plausible than a normal fibroblast being the terminal state. In my opinion, the pseudotime analyses should be scrapped entirely, or if included, either the corresponding manuscript text needs to be re-written to reflect more uncertainty and the need for future studies, or additional validation/analysis is required. One potential way to make this data more convincing is to perform the same trajectory analyses on NS fibroblasts and show a lack of continuum or different results from PP. It seems a similar trajectory could be constructed with NS fibroblasts and show a similar result as PP fibroblasts, especially since most of these cells are present in NS, unless authors can prove this wrong.

We thank the reviewer for these comments. We agree that the pseudotime trajectory for the PP fibroblasts may be somewhat arbitrary. Unlike the keratinocyte pseudotime trajectories that were constructed based on reasonable assumptions that epidermal cells differentiate from the basal layer to the outer layer, the fibroblast trajectory may not have been built from equally robust assumptions. Thus, we have removed the fibroblast pseudotime analysis from the manuscript per the reviewer's suggestion. As an alternative, we focused on the heterogeneity of the fibroblasts and emphasized the highly inflammatory characteristics of cluster 6 in PP skin.

2. Additional IF staining in Extended Fig. 9 appears to lack the same number of co-stained cells as those from Main Fig. 4 and no other areas besides the upper dermis to demonstrate specificity of the localization of these cells. This suggests either the SFRP2+ cells are in other areas of the dermis or there was a technical issue with the IF (e.g., poor antibody), and fails to support their assertion that there are abundant SFRP2+ fibroblasts in the upper dermis near epidermis in PP lesions that produce these chemokines and CTSS. As presented, the IF data suggest that other cell types in the upper dermis (such as myeloid cells) appear to be the dominant source of these chemokines/CTSS.

We thank the reviewer for these comments. We agree that fibroblasts are not the only source of these cytokines in psoriatic skin and have revised our text to address that. Sometimes these types of staining can be finicky and the IF for SFRP2 showed positive but weak staining – we have repeated this staining as shown in **Figure R4**, which now replaces the old Extended Data Fig. 9b. In terms of the localization of the SFRP2 positive fibroblasts being in the upper dermis, we additionally provide single-stain IHC shown in Extended Figure 10a.

[Editorial note: Figure R4 redacted]

REVIEWER COMMENTS

Reviewer #2 (Remarks to the Author):

The authors have responded relatively well to the second round of reviews. However, the concern raised in both review 1 and 2 regarding their claim that 'CD8+ Tc17 are the major source of IL17a in psoriatic skin" has not been adequately addressed. CD4 T cells are now included in scRNA-seq clustering which is helpful. I do not disagree that their data shows a limited role for CD4. However, in order to move the field away from the well-established concept that CD4 are a key source of IL17 in Psx, this point needs to be directly addressed. This demonstration will increase the impact of the manuscript. A simple IF image showing CD4 vs CD8 and IL-17 in the skin along with controls would support their claim. Note that Figure Supplemental 11b demonstrates feasibility of this approach but is difficult to interpret due to high background. Showing skin stained with anti-CD3, Anti-cd4 or anti-CD8, and anti-IL17A would address this questions. Please show single color and overlaid images to allow for interpretation. This will address potential differences between mRNA and protein. Please also discuss this point in the discussion section. Lacking these additional data, this claim (including section heading and inclusion in abstract) should be reduced.

Reviewer #3 (Remarks to the Author):

I thank the authors for considering my suggestions and addressing my concerns, which they have done adequately.

Reviewer #2 (Remarks to the Author):

The authors have responded relatively well to the second round of reviews. However, the concern raised in both review 1 and 2 regarding their claim that "CD8+ Tc17 are the major source of IL17a in psoriatic skin" has not been adequately addressed. CD4 T cells are now included in scRNA-seq clustering which is helpful. I do not disagree that their data shows a limited role for CD4. However, in order to move the field away from the well-established concept that CD4 are a key source of IL17 in Psx, this point needs to be directly addressed. This demonstration will increase the impact of the manuscript. A simple IF image showing CD4 vs CD8 and IL-17 in the skin along with controls would support their claim. Note that Figure Supplemental 11b demonstrates feasibility of this approach but is difficult to interpret due to high background. Showing skin stained with anti-CD3, Anti-cd4 or anti-CD8, and anti-IL17A would address this questions. Please show single color and overlaid images to allow for interpretation. This will address potential differences between mRNA and protein. Please also discuss this point in the discussion section. Lacking these additional data, this claim (including section heading and inclusion in abstract) should be reduced.

Response:

We appreciate the comments from the reviewer and the chance to respond to them. While our findings described here suggest that CD8⁺ T cells are a major source of IL-17A it does not mean that IL-17A from CD4 T cells is not important in the pathogenesis, and we like to highlight that we also do identify IL-17A coming from CD4⁺ T cells as shown in our data (See Figure 5D in manuscript). As mentioned in our manuscript there has been growing evidence for IL-17 producing CD8⁺ cells in psoriasis (see refs 33-35 in our manuscript), and a recent publication by Paola Di Meglio *et al.* demonstrated that targeting CD8⁺ T cells prevents psoriasis development (see PMID:26782974), and a majority of the CD8⁺ T cells in that model expressed IL-17A. Similarly in a publication by Pieter C. M. Res IL-17A-producing CD8⁺ T cells were found at a higher proportion than CD4⁺ T cells (ref 35). These data are in alignment with the findings presented in our manuscript. To try to validate these findings better we performed IF staining of lesional psoriatic skin using anti-IL-17A and anti-CD4 or anti-CD8 separately (we have had difficulties doing 3-color or 4-color IFs). Three representative panels for IL-17A/CD4 and IL-17A/CD8 are shown in Figure R1 in response to the reviewer's comments.

We have further analyzed our single-cell data to determine the contribution of CD4⁺ and CD8⁺, and what fraction of CD8⁺ T cells express IL-17A. As shown in Figure R2 only a portion of CD8⁺ T cells in psoriatic skin expresses IL17A, consistent with our findings from the IF staining. In addition, the expression of IL23R is mostly restricted to this same subset in cluster #5 (See Extended Data Fig. 11b), aligning with this cluster being a major source of IL17A in psoriasis. However, we would like to emphasize that the focus of our manuscript is to provide a comprehensive overview of the single-cell landscape of psoriasis with an emphasis on the stromal cell populations such as keratinocytes and fibroblasts, of which fibroblasts have not been well studied. While, we do not feel that shifting the focus of our manuscript to T-cells, which have been extensively studied in psoriasis as mentioned above, is warranted, we have taken steps towards adjusting the text of our manuscript.

Figure R1. Immunofluorescence for IL-17A/CD4 (top three IF panels) and IL-17A/CD8 (bottom three IF panels, 10X Magnification). Right insert, Quantification of IL-17A⁺ vs IL-17A⁻ cells for both CD4 and CD8 cells. While we found substantial number of IL-17A⁺ cells in psoriatic skin, we observed limited number of CD4⁺ T cells. In contrast we found greater number of CD8⁺ T cells. As calculated for each slide IL17A/CD4 and IL17A/CD8 (15 slides in total, 4 for IL-17A/CD4 and 11 IL-17A/CD8), the proportion of IL-17A producing CD4⁺ T cells was high (up to 40%) compared to 20% of CD8⁺ T cells per 10X magnification. However, as there were relatively low number of IL-17A⁺CD4⁺ T cells per 10X magnification compared to the total number of IL-17A⁺CD8⁺ T cells, suggesting that CD8⁺ T cells are a larger contributor to the total IL-17A production compared to CD4⁺ T cells, which also is consistent with our single-cell findings (see Supplemental Figure 11c).

[Editorial note: Figure R1 redacted]

Thus, in the results sections, instead of stating that “CD8+ Tc17 are **the** major source of IL17A in psoriatic skin” we have changed this to “CD8+ Tc17 are **a major** source of IL17A in psoriatic skin”. We also added the new plots in Figure R1 and R2 to **Extended Data Fig. 12**.

Figure R2. Additional analyses from the single-cell data. This shows the co-localization of gene expression across all the T cells (A). Most of the IL-17A was found in a single-cluster of T cells, cluster #5, however, this only represented a small proportion of the overall CD8A⁺ T cells in psoriatic skin.

[Editorial note: Figure R2 redacted]

Reviewer #3 (Remarks to the Author):

I thank the authors for considering my suggestions and addressing my concerns, which they have done adequately.

Response: We appreciate that the reviewer is satisfied with our revision.

REVIEWERS' COMMENTS

Reviewer #2

Noted in private comments that concerns adequately addressed.